# Nrf1 promotes heart regeneration and repair by regulating proteostasis and redox balance

Miao Cui[1,2], Ayhan Atmanli[1,2], Maria Gabriela Morales[1,2], Wei Tan[1,2], Kenian Chen [3], Xue Xiao[3], Lin Xu [3], Ning Liu [1,2], Rhonda Bassel-Duby [1,2] & Eric N. Olson [1,2✉]

Following injury, cells in regenerative tissues have the ability to regrow. The mechanisms whereby regenerating cells adapt to injury-induced stress conditions and activate the regenerative program remain to be defined. Here, using the mammalian neonatal heart regeneration model, we show that Nrf1, a stress-responsive transcription factor encoded by the Nuclear Factor Erythroid 2 Like 1 (*Nfe2l1*) gene, is activated in regenerating cardiomyocytes. Genetic deletion of *Nrf1* prevented regenerating cardiomyocytes from activating a transcriptional program required for heart regeneration. Conversely, Nrf1 overexpression protected the adult mouse heart from ischemia/reperfusion (I/R) injury. Nrf1 also protected human induced pluripotent stem cell-derived cardiomyocytes from doxorubicin-induced cardiotoxicity and other cardiotoxins. The protective function of Nrf1 is mediated by a dual stress response mechanism involving activation of the proteasome and redox balance. Our findings reveal that the adaptive stress response mechanism mediated by Nrf1 is required for neonatal heart regeneration and confers cardioprotection in the adult heart.

---

[1] Department of Molecular Biology, the Hamon Center for Regenerative Science and Medicine, University of Texas Southwestern Medical Center, Dallas, TX, USA. [2] Senator Paul D. Wellstone Muscular Dystrophy Specialized Research Center, University of Texas Southwestern Medical Center, Dallas, TX, USA. [3] Quantitative Biomedical Research Center, Department of Population & Data Sciences and Department of Pediatrics, University of Texas Southwestern Medical Center, Dallas, TX, USA. ✉email: eric.olson@utsouthwestern.edu

Heart disease represents a leading cause of morbidity and mortality in the developed world. Common types of heart disease include coronary artery disease, hypertension, and chemotherapy, all of which lead to loss or dysfunction of cardiac muscle. Preventing cardiomyocytes from death (cardioprotection) and replenishing the lost myocardium (regeneration) are the central goals for heart repair.

The adult mammalian heart lacks the ability to regenerate and, instead, responds to injury by replacing lost tissue with a non-contractile fibrotic scar, resulting in pathological remodeling that can eventually lead to heart failure[1,2]. In contrast, the heart of neonatal mice, within the first week after birth, possesses the ability to regenerate lost myocardium, mediated by the proliferation of cardiomyocytes[3,4]. The molecular mechanisms underpinning the neonatal regenerative response and its absence in later life remain to be defined. Using single-nucleus RNA sequencing (snRNA-seq), we previously showed that neonatal regenerating cardiomyocytes activate a unique transcriptional response upon injury[5]. Interestingly, besides cell-cycle activation, this regenerative cell population also upregulates cell survival pathways, suggesting a coregulation between cardioprotection and heart regeneration. Although cardioprotection and heart regeneration have been traditionally thought to involve separate mechanisms, protection of cardiomyocytes from injury or disease stimuli is a prerequisite to any meaningful regenerative response.

Here, we sought to study how neonatal cardiomyocytes cope with injury-induced stress to regenerate damaged myocardium and whether the underlying mechanisms could be leveraged to promote heart regeneration and repair in adulthood. By spatial transcriptomic profiling, we visualized regenerative cardiomyocytes reconstituting damaged myocardium after ischemic injury and found that they are marked by the expression of Nrf1, an ER-bound stress-responsive transcription factor. SnRNA-seq revealed that genetic deletion of Nrf1 prevented neonatal cardiomyocytes from activating a transcriptional program required for heart regeneration. Conversely, overexpression of Nrf1 using AAV9 gene therapy protects the adult mouse heart from ischemia/reperfusion (I/R) injury. The protective function of Nrf1 is mediated by the proteasomal activity and antioxidant response, including activation of the detoxifying enzyme Hmox1. Nrf1 also protects human-induced pluripotent stem cell-derived cardiomyocytes (hPSC-CMs) from cardiotoxicity induced by the chemotherapeutic drug doxorubicin (Dox) and other toxic agents. Taken together, our study uncovers a unique adaptive mechanism activated in response to injury that maintains the tissue homeostatic balance required for heart regeneration. Reactivating this mechanism in the adult heart represents a potential therapeutic approach for cardiac repair.

## Results

### The spatial transcriptome of the regenerating mouse heart.
Regenerating cardiomyocytes in neonatal hearts are derived from preexisting cardiomyocytes instead of cardiac stem cells[3,4,6]. Using snRNA-seq, we previously identified a population of cardiomyocytes (termed CM4) in the neonatal regenerative heart that enters the cell cycle after injury[5]. CM4 cells exhibited an embryonic gene signature marked by the expression of Acta2 (Fig. 1a)[5], a smooth muscle actin isoform that is transiently expressed in the embryonic heart[7]. Using an ultrasensitive and semi-quantitative RNA detection method (RNA-scope)[8], we found that ~30% of cardiomyocytes in the regenerative heart at postnatal day (P) 1 showed enriched Acta2 expression (Fig. 1b, c). This group of high Acta2-expressing cardiomyocytes was not detected in nonregenerative hearts at P8 (Fig. 1c, d), validating our previous observation that CM4 cells disappear as the heart loses its ability to regenerate[5].

To visualize CM4 cells during heart regeneration, we analyzed the spatial transcriptome of heart sections collected at 3 and 7 days after myocardial infarction (MI) or Sham surgery performed at P1. Heart sections at 200 µm below the ligation of the left anterior descending artery (LAD) in MI hearts or the equivalent position in sham hearts were used for the analysis. Using the Visium platform offered by 10× Genomics, we captured on average 672 spatial spots for each section and 5549 detected genes per spot (Supplementary Fig. 1a). Because each spot captures multiple cells, to deconvolute the underlying composition of cell types we integrated the spatial transcriptome data with our previous snRNA-seq datasets using a computational approach that identifies shared cell states (see the "Methods" section)[9]. The resulting data calculated the fraction of each cell type per given spot and mapped the anatomic distribution of cardiac cell types onto heart sections, which was further corroborated by the spatial expression of marker genes (Fig. 1e–g and Supplementary Fig. 1b and c)[10]. Our analysis correctly identified the known anatomic localization of endocardial cells (Fig. 1h) and revealed a transient immune cell infiltration and accumulation of fibroblasts in the infarct region at day-3 and day-7 post-MI, respectively (Fig. 1i, j and Supplementary Fig. 2), consistent with our previous studies[3,11]. CM1 cells, the most abundant cardiomyocyte population, were detected across the entire myocardium (Fig. 1k), whereas CM5 cells, a group of injury-induced cardiomyocytes, were localized in the border zone adjacent to the injured tissue (Fig. 1l), consistent with previous immunohistochemistry results[5]. Importantly, we found that, while CM4 cells were sporadically localized in the Sham heart (Supplementary Fig. 2), they became enriched in the infarct region and were proximal to the fibrotic tissue in MI hearts (Fig. 1m and Supplementary Fig. 2). The localization of CM4 cells in the infarct myocardium that is undergoing regrowth, along with their immature and proliferative transcriptional signatures[5], provide strong evidence that they represent regenerating cardiomyocytes that reconstitute the damaged myocardium.

### Regenerative cardiomyocytes express high levels of Nrf1.
Our previous analysis showed that, besides cell-cycle activation, CM4 cells also upregulate cell survival pathways following ischemic injury[5]. Indeed, protein poly-ubiquitination and proteasomal activity are maintained in the P1 regenerative heart after MI (Supplementary Fig. 3), indicating preserved proteostasis during heart regeneration. This contrasts with the P8 non-regenerative heart, which showed increased ubiquitinated proteins and decreased proteasomal activity after MI, reflecting proteolytic stress (Supplementary Fig. 3).

Recent studies showed that Nrf1 (encoded by the Nfe2l1 gene) functions as a guardian of tissue homeostasis by regulating proteostasis and redox balance in response to cellular stress[12–14]. Nrf1 belongs to the Cap'N'Collar (CNC) family of leucine zipper transcription factors, which include Nrf1, Nrf2, Nrf3, and Nfe2[15,16], and is the most highly expressed member in the heart with enriched expression in cardiomyocytes (Supplementary Fig. 4a, b). However, despite the high abundance, its cardiac functions are entirely unknown. From our previous snRNA-seq data, we found that Nrf1 expression is enriched in CM4 cardiomyocytes (Supplementary Fig. 4c) and is upregulated after MI[5]. Further analysis showed that the expression profile of Nrf1 correlates with the expression of the CM4 marker Acta2, as well as with the cell-cycle genes Mki67 and Ccnb2, in individual cardiomyocytes (Fig. 1n and Supplementary Fig. 4d). Using RNA-scope, we verified that Nrf1 transcripts are co-expressed with Acta2 in CM4 cells (Supplementary Fig. 4e, f) and are expressed at higher levels in P1 compared to P8 cardiomyocytes (Fig.1o–q), suggesting a potential role of Nrf1 in neonatal heart regeneration.

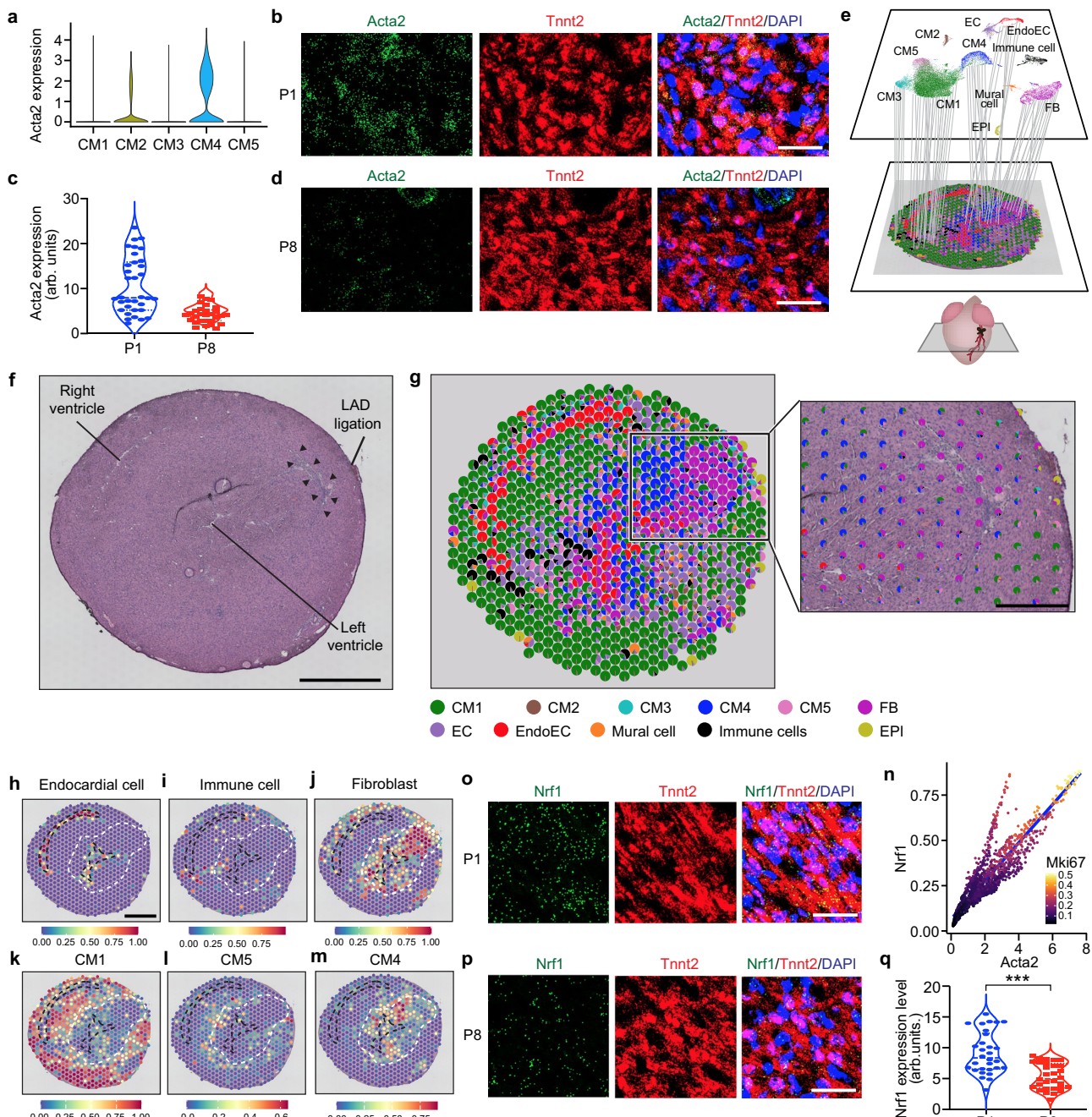

**Fig. 1 Nrf1 is expressed in regenerative CM4 cardiomyocytes. a** Violin plot showing the expression of *Acta2* in CM1–CM5 cardiomyocyte populations. Expression levels are shown as reads per 10k counts. **b** *Acta2* and *Tnnt2* RNA transcripts detected by RNA-scope probes in the left ventricular myocardium on a transverse section of P1 hearts. Scale bar, 20 μm. **c** Quantification of *Acta2* transcript levels in cardiomyocytes from P1 and P8 heart; n = 29–34 cardiomyocytes were quantified. **d** *Acta2* and *Tnnt2* RNA transcripts detected by RNA-scope probes in the left ventricular myocardium on a transverse section of P8 hearts. Scale bar, 20 μm. **e** Schematic diagram showing the spatial transcriptome analysis of heart sections to identify anatomic locations of cardiac cell type populations. **f** Hematoxylin and eosin (H&E) image of a heart transverse section collected at 7-day post P1 MI. Left and right ventricles are indicated. Black arrowheads depict fibrotic tissue; Scale bar, 500 μm. **g** Left: cell type composition shown in pie-charts for individual spatial spots mapped onto the heart section with H&E staining shown in (**f**). Right: the boxed region is shown in high magnification overlayed with the H&E image to highlight the fibrotic region; Scale bar, 200 μm. **h–m** Anatomic localization of endocardial cells (**h**), immune cells (**i**), fibroblasts (**j**), CM1 (**k**), CM5 (**l**), and CM4 (**m**) on the same heart section in (**f**). Color scale indicates cell type ratio from 0 to 1. Endocardium is outlined by black dash lines, and infarct region, which is identified by the location of suture, is outlined in white; Scale bar, 500 μm. **n** Expression correlation plots between *Nrf1*, *Acta2*, and *Mki67* in individual cardiomyocytes from our previous snRNA-seq data (GSE130699). *Nrf1* positively correlates with *Acta2* (Spearman's coefficient = 0.91). Color scale represents the expression level of *Mki67*. Expression levels are shown as reads per 10k counts. **o, p** *Nrf1* and *Tnnt2* RNA transcripts detected by RNA-scope probes in the left ventricular myocardium on transverse sections of P1 hearts (**o**) and P8 hearts (**p**). Scale bar, 20 μm. **q** Quantification of *Nrf1* expression levels in cardiomyocytes from P1 and P8 hearts; n = 30 cardiomyocytes were quantified for each group. **f–m** experiments were repeated independently twice with similar results. ***p < 0.0001, Student's t-test two-tailed. **c** and **q** arb. units. arbitrary units. **e** and **g** EC endothelial cells, EndoEC endocardial cells, FB fibroblasts, EPI epicardial cells.

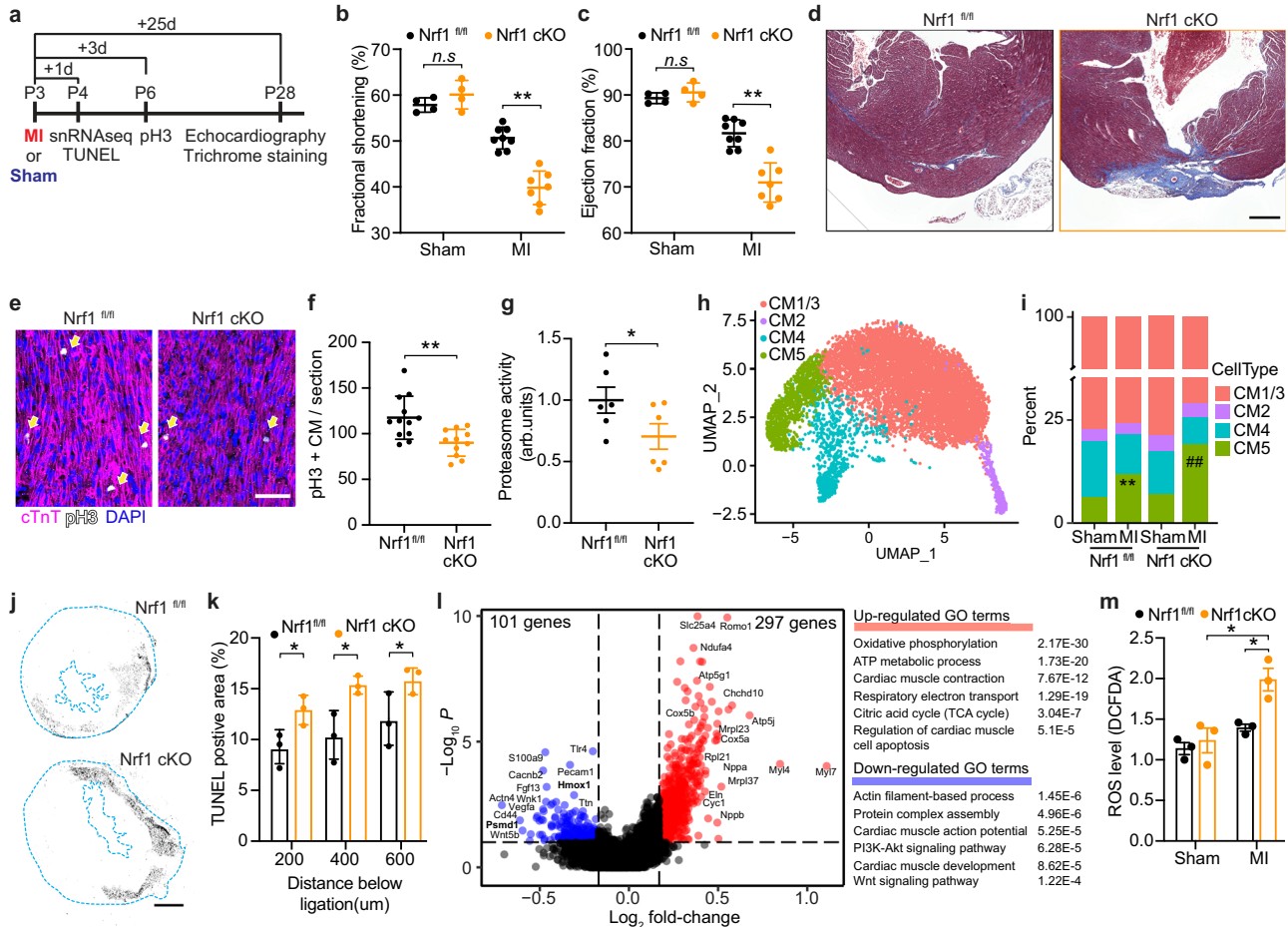

**Fig. 2 Nrf1 deletion impairs neonatal heart regeneration. a** Illustration showing timeline of MI and Sham surgeries performed at P3 and subsequent days of sample collection. **b**, **c** Fractional shortening (**b**) and ejection fraction (**c**) of Nrf1$^{fl/fl}$ and Nrf1 cKO mouse hearts at 25 days after MI or Sham surgery; $n = 4$ animals for Sham groups and $n = 7$–8 animals for MI groups; **$p = 0.0022$ (**b**), **$p = 0.0009$ (**c**) by Student's $t$-test two-tailed. **d** Masson's trichrome staining showing fibrotic scarring in Nrf1$^{fl/fl}$ and Nrf1 cKO mouse hearts at 25-day after MI; similar results were observed in six animals; Scale bar, 200 μm. **e** Immunohistochemistry showing pH3$^+$ cardiomyocytes (cTnT$^+$), marked by arrows, in heart sections collected from Nrf1 cKO and Nrf1$^{fl/fl}$ mice 3-day post-MI; Scale bar, 50 μm. **f** Quantification of pH3$^+$ cardiomyocytes (cTnT$^+$) in heart sections from (**e**); $n = 12$ animals for WT and $n = 11$ animals for Nrf1 cKO, **$p = 0.0034$ by Student's $t$-test two-tailed. **g** Chymotrypsin-like activity of the proteasome in hearts from Nrf1 cKO and Nrf1$^{fl/fl}$ mice at P4. arb. units, arbitrary units; $n = 6$ animals for each group; *$p = 0.0373$ by Student's $t$-test one-tailed. **h** UMAP visualization of cardiomyocyte clusters colored by identity. CM1 and CM3 were merged as a single cluster due to their transcriptome similarity. **i** Fraction of cardiomyocyte populations in Nrf1$^{fl/fl}$ and Nrf1 cKO hearts in MI or Sham conditions; $n = 4$–6 animals for each group examined over 1 independent experiment; **$p = 4.49E^{-16}$ compared to WT-Sham, ##$p = 5.05E^{-31}$ compared to KO-Sham; Chi-squared test. **j** Representative images showing TUNEL staining on heart sections from Nrf1 cKO and Nrf1$^{fl/fl}$ mice at 1-day post-MI; Scale bar, 500 μm. **k** Quantification of TUNEL positive area in heart sections collected at 200, 400, and 600 μm below the ligation from Nrf1 cKO and Nrf1$^{fl/fl}$ mice at 1-day post-MI; $n = 3$ animals for each group; *$p = 0.039$ (200 μm); $p = 0.025$ (400 μm), $p = 0.047$ (600 μm), Student's $t$-test two-tailed. **l** Volcano plot showing fold-change and $p$-value of genes up-regulated (red) and down-regulated (blue) in CM4 cells of Nrf1 KO hearts compared to CM4 cells in control hearts at 1-day post-MI; $p$-adjust < 0.1 and fold-change >1.5 were used for cutoffs; $p$-adjust values are calculated by Wilcoxon rank-sum test two-tailed. Top enriched GO terms are shown (right). **m** Quantification of ROS levels in Nrf1 cKO and Nrf1$^{fl/fl}$ mice at 3-day post-MI or Sham; $n = 3$ animals for each group. WT-MI vs Nrf1 cKO-MI, *$p = 0.015$; Nrf1 cKO-Sham vs. Nrf1 cKO-MI, *$p = 0.022$, by Student's $t$-test two-tailed. **b**, **c**, **f**, **g**, **k**, **m** results are shown as mean ± s.e.m.; n.s. not significant.

**Nrf1 is required for neonatal heart regeneration.** Mice with a cardiomyocyte-specific knockout of *Nrf1* (Nrf1 cKO), generated by crossing Nrf1$^{fl/fl}$ mice with αMHC-Cre transgenic mice, exhibited no aberrations in cardiac morphology or function at 2-month of age (Supplementary Fig. 5a, b). To determine whether Nrf1 was required for neonatal heart regeneration, we performed MI on Nrf1 cKO and Nrf1$^{fl/fl}$ (control) mice at P3, followed by echocardiography and histological analyses at P28 (Fig. 2a). While control hearts regenerated as expected, Nrf1 cKO hearts showed decreased cardiac function, measured as fractional shortening (FS), ejection fraction (EF), and myocardial wall motion (Fig. 2b, c, and Supplementary Fig. 5c), as well as

increased fibrotic scarring (Fig. 2d and Supplementary Fig. 5d). Cardiomyocyte proliferation at 3-day post-MI was also significantly decreased in Nrf1 cKO hearts (Fig. 2e, f), underlying their impaired heart regeneration. Additionally, proteasomal activity was notably reduced in Nrf1 cKO hearts (Fig. 2g), consistent with the role of Nrf1 in regulating the proteasome[17].

Neonatal cardiomyocytes are heterogeneous and comprised of five subpopulations (CM1–CM5)[5]. To understand which cardiomyocyte population(s) account for the impaired heart regeneration in Nrf1 cKO mice, we performed snRNA-seq on cardiomyocytes from Nrf1 cKO and control hearts at 1-day post-MI or Sham surgery performed at P3. We sequenced 8585

cardiomyocytes and annotated the subpopulations based on the expression of their marker genes (Fig. 2h and Supplementary Fig. 6)[5]. While the Nrf1 cKO and control hearts had comparable cardiomyocyte composition in the Sham samples, after MI, the Nrf1 cKO hearts showed a significant increase of CM5 cells—the cardiomyocytes that respond to injury by apoptosis and hypertrophy[5] (Fig. 2i). The enhanced induction of CM5 cells in the Nrf1 cKO hearts suggested increased cell death after MI. Indeed, TUNEL staining on heart samples collected at 1-day post-MI revealed a greater myocardial loss in the Nrf1 cKO hearts compared to controls (Fig. 2j, k).

To understand how *Nrf1* deletion affects the ability of CM4 cells to regenerate, we performed differentially expressed gene (DEG) analysis on CM4 cells in the MI samples from Nrf1 cKO and control hearts (Supplementary Data 1). We found that the oxidative phosphorylation (OXPHOS) pathway, which is down-regulated in wild-type (WT) CM4 cells during heart regeneration[5], was instead upregulated in Nrf1 cKO hearts (Fig. 2l). Direct measurement of reactive oxygen species (ROS) by 2′,7′-dichlorodihydrofluorescein diacetate (DCFDA) oxidation assay showed a significant increase of ROS in Nrf1 cKO MI hearts compared to control MI hearts (Fig. 2m). In addition, *Nppa* and *Nppb*, markers of cardiac stress, were both upregulated by *Nrf1* deletion. Down-regulated genes in Nrf1 cKO hearts encode proteins that are associated with actin filament-based processes, the PI3K–AKT signaling pathway, and the Wnt signaling pathway, and included known Nrf1 target genes, such as *Hmox1* and *Psmd1*, which encode a detoxifying enzyme and a proteasome subunit, respectively (Fig. 2l)[18]. Notably, both the actin filament-based processes and the PI3K–AKT signaling pathway were activated in CM4 cells of WT hearts during heart regeneration[5]. Therefore, *Nrf1* deletion increased OXPHOS metabolism, which is known to inhibit cardiomyocyte proliferation[19], and prevented CM4 cells from initiating the transcriptional response required for heart regeneration.

**Nrf1 overexpression protects adult hearts from I/R injury.** To study whether Nrf1 overexpression might protect adult hearts against ischemia/reperfusion (I/R) injury, we generated cardio-trophic adeno-associated viral (AAV) vectors that expressed Nrf1 (AAV9-Nrf1) or TdTomato (control) (Fig. 3a and Supplementary Fig. 7a)[20,21]. We ligated the LAD artery for 45 min followed by 24 h of reperfusion on 8-week-old mice pre-injected with AAV9 at P4 (Fig. 3b). We analyzed infarct size at the end of reperfusion by 2,3,5-Triphenyltetrazolium chloride (TTC) staining and cardiac function on three consecutive weeks after the surgery. At the baseline level, AAV-Nrf1 hearts had similar cardiac function and proportions of cardiac fibroblasts as well as tissue-resident immune cells compared to AAV9-TdTomato hearts (Supplementary Fig. 7b–e). However, after injury, AAV9-Nrf1 significantly reduced the infarct area from 52% to 40% (Fig. 3c, d) and improved FS and EF after I/R (Fig. 3e, f). Additionally, the increase of diastolic and systolic volume of the left ventricle after MI, as seen in the control hearts, was profoundly attenuated in AAV9-Nrf1 treated mice (Fig. 3g and Supplementary Fig. 8a), which showed reduced cardiac dilation and remodeling. Masson's trichrome staining at 3 weeks post-IR further revealed reduced fibrotic scarring in AAV9-Nrf1 mice (Fig. 3h, i and Supplementary Fig. 8b). Together, these data indicate that overexpression of Nrf1 in vivo confers a protective effect in adult hearts following ischemic injury.

Next, to determine whether Nrf1 overexpression confers a broad protective benefit against general cardiac insults, we treated neonatal rat ventricular myocytes (NRVMs) overexpressing Nrf1 or LacZ (control) with $H_2O_2$ (an oxidative stress inducer),

peroxynitrite (a reactive nitrogen species released during I/R)[22], Dox (a cardiotoxic chemotherapeutic drug)[23], or erastin (a ferroptosis inducer)[24], followed by analyses of cell survival. We observed a consistent protective effect of Nrf1 overexpression in the presence of all of these cardiotoxins, as revealed by Calcein AM staining that labels live cells (Fig. 3j). Nrf2, which is a family member of Nrf1, is a stress regulator that also confers cardio-protection[25]. Despite similar overexpression levels (Supplementary Fig. 9a), the protective effect of Nrf1 was greater than that of Nrf2 in response to all these cardiotoxin treatments (Fig. 3k), highlighting the importance of Nrf1 as a key regulator of cellular stress in the heart.

**The transcriptional activity of Nrf1 is required for cardioprotection.** Nrf1 is an ER-bound transcription factor that is translocated to the nucleus upon the cleavage near the N-terminus. We observed increased nuclear translocation of Nrf1 in NRVMs after $H_2O_2$ treatment (Supplementary Fig. 10a), suggesting that its nuclear function is involved in cytoprotection. To uncouple the transcriptional function of Nrf1 from its potential function in the ER, we generated truncation mutants of Nrf1 lacking the ER-binding domain within amino acids 1–103 (Nrf1-ΔN) or the DNA-binding domain within amino acids 608–741 (Nrf1-ΔC). Nrf1-ΔN corresponds to the naturally cleaved nuclear form of Nrf1, whereas Nrf1-ΔC displayed similar localization to Nrf1 (Supplementary Fig. 10b). Despite similar levels of overexpression (Supplementary Fig. 10c), Nrf1-ΔN conferred marked protection against $H_2O_2$, whereas Nrf1-ΔC was not protective (Supplementary Fig. 10d, e), showing that the transcriptional activity of Nrf1 is required for its cytoprotective function.

**Nrf1 regulates proteostasis and redox balance.** To study the transcriptional program underlying the protective function of Nrf1, we performed RNA sequencing on NRVMs overexpressing LacZ, Nrf1, or Nrf2. DEG analysis between Nrf1 and LacZ samples identified 1241 genes that were upregulated by Nrf1 (Supplementary Data 2). Gene Ontology (GO) analysis revealed that up-regulated genes were associated with proteasome-mediated proteolysis, such as the proteasome protein catabolism and ER-associated degradation (ERAD) pathways, as well as the antioxidant response, such as glutathione and nicotinamide adenine dinucleotide phosphate (NADPH) metabolism (Fig. 4a). Consistent with upregulation of the antioxidant response, oxidative DNA damage, marked by 8-Oxo-2′-deoxyguanosine (8-OHdG), and activation of cellular, as well as mitochondrial ROS following $H_2O_2$ treatment, were significantly attenuated in Nrf1-expressing cells compared to control cells (Supplementary Fig. 11). While pathways related to the antioxidant response were activated by both Nrf1 and Nrf2, the activation of proteasome-mediated proteolysis pathways was unique to Nrf1 over-expression (Fig. 4b, c). Consistent with this result, all three major cleavage activities of the proteasome (trypsin-like, chemotrypsin-like, and caspase-like) were significantly increased in NRVMs overexpressing Nrf1, which was not observed in response to Nrf2 overexpression (Fig. 4d). In addition, genes related to the innate immune response and cytokine production were down-regulated in both Nrf1 and Nrf2 samples, suggesting a shared anti-inflammatory function, whereas genes involved in extracellular matrix organization were specifically down-regulated by Nrf1 (Supplementary Fig. 9b).

Proteasome subunit genes, *Psma1*, *Psmd1*, and *Psmb3*, as well as antioxidant genes, *Hmox1*, *Sod1*, and *Cat*, were downregulated in Nrf1 cKO hearts and upregulated in AAV9-Nrf1 hearts, suggesting that they are direct targets of Nrf1 and underlie its cardioprotective function (Fig. 4e, f). To study potential

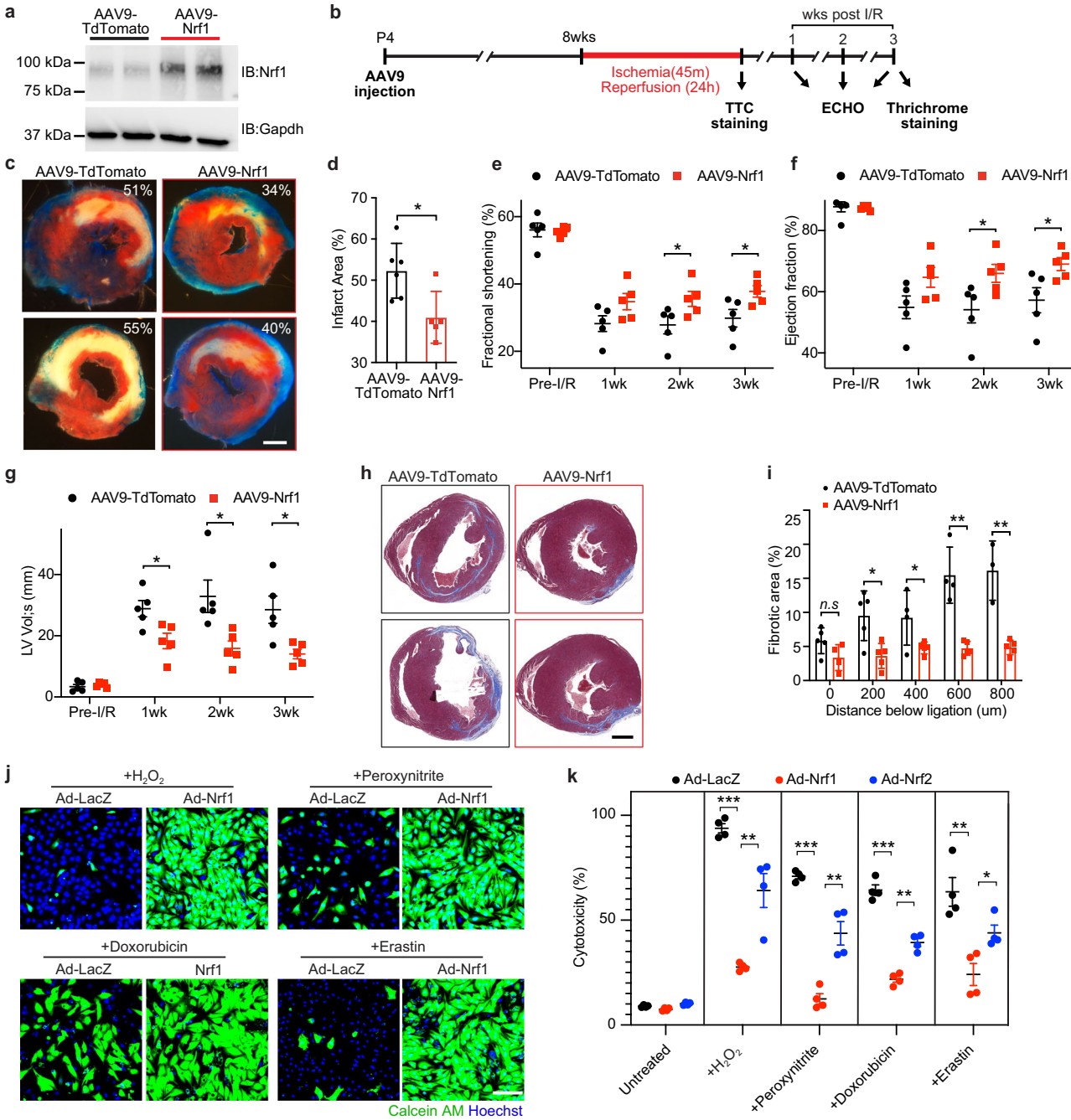

**Fig. 3 Nrf1 overexpression protects cardiomyocytes against I/R and cardiotoxin injury. a** Western blot analysis showing Nrf1 expression in hearts from mice at 4-week after AAV9-TdTomato and AAV9-Nrf1 injection at P4; experiments were repeated independently twice with similar results. **b** Illustration of a timeline showing the experimental design for AAV9 injection and time points of sample collections. **c** TTC staining showing infarct size on transverse heart sections collected at the end of reperfusion; Scale bar, 500 μm. **d** Quantification of infarct area in sections from (**c**); AAV9-TdTomato, $n = 6$; AAV9-Nrf1, $n = 5$; *$p = 0.0183$, by Student's $t$-test two-tailed. **e–g** Fractional shortening (**e**), ejection fraction (**f**), and systolic volume (**g**) of left ventricles from AAV9-TdTomato and AAV9-Nrf1 hearts at baseline (before I/R), and 1–3 weeks (wk) after I/R; $n = 5$ animals for each group. **e** 2wk, *$p = 0.0388$; 3wk, *$p = 0.034$. **f** 2wk, *$p = 0.0422$; 3wk, *$p = 0.0336$. **g** 1wk, *$p = 0.0207$, 2wk, *$p = 0.0188$; 3wk, *$p = 0.0155$. **h** Masson's trichrome staining showing fibrotic scarring on transverse sections of hearts collected at 600 μm below the ligation suture; Scale bar, 500 μm. **i** Quantification of fibrotic regions of heart sections collected at multiple planes below the ligation suture from AAV-TdTomato and AAV-Nrf1 mice; AAV9-TdTomato, $n = 5$; AAV9-Nrf1, $n = 4$; 200 μm, *$p = 0.012$, 400 μm, *$p = 0.0455$, 600 μm, *$p = 0.0007$, 800 μm, *$p = 0.0012$. **j** Viable NRVMs with adenoviral (Ad) overexpression of LacZ or Nrf1 following $H_2O_2$, peroxynitrite, doxorubicin, and erastin treatments, indicated by Calcein AM (green); Scale bar, 100 μm. **k** Cell death measured by lactate dehydrogenase (LDH) release in NRVMs-overexpressing LacZ, Nrf1, or Nrf2 following toxin treatments; $n = 4$ biologically independent experiments for each group. $H_2O_2$, ***$p < 0.000001$, **$p = 0.0043$. Peroxynitrite, ***$p < 0.000001$, **$p = 0.0022$. Doxorubicin, ***$p = 0.000006$, **$p = 0.0003$. Erastin, **$p = 0.0039$, *$p = 0.0215$. **d–g**, **i**, **k** Results are shown as mean ± s.e.m.; Student's $t$-test two-tailed; n.s. not significant.

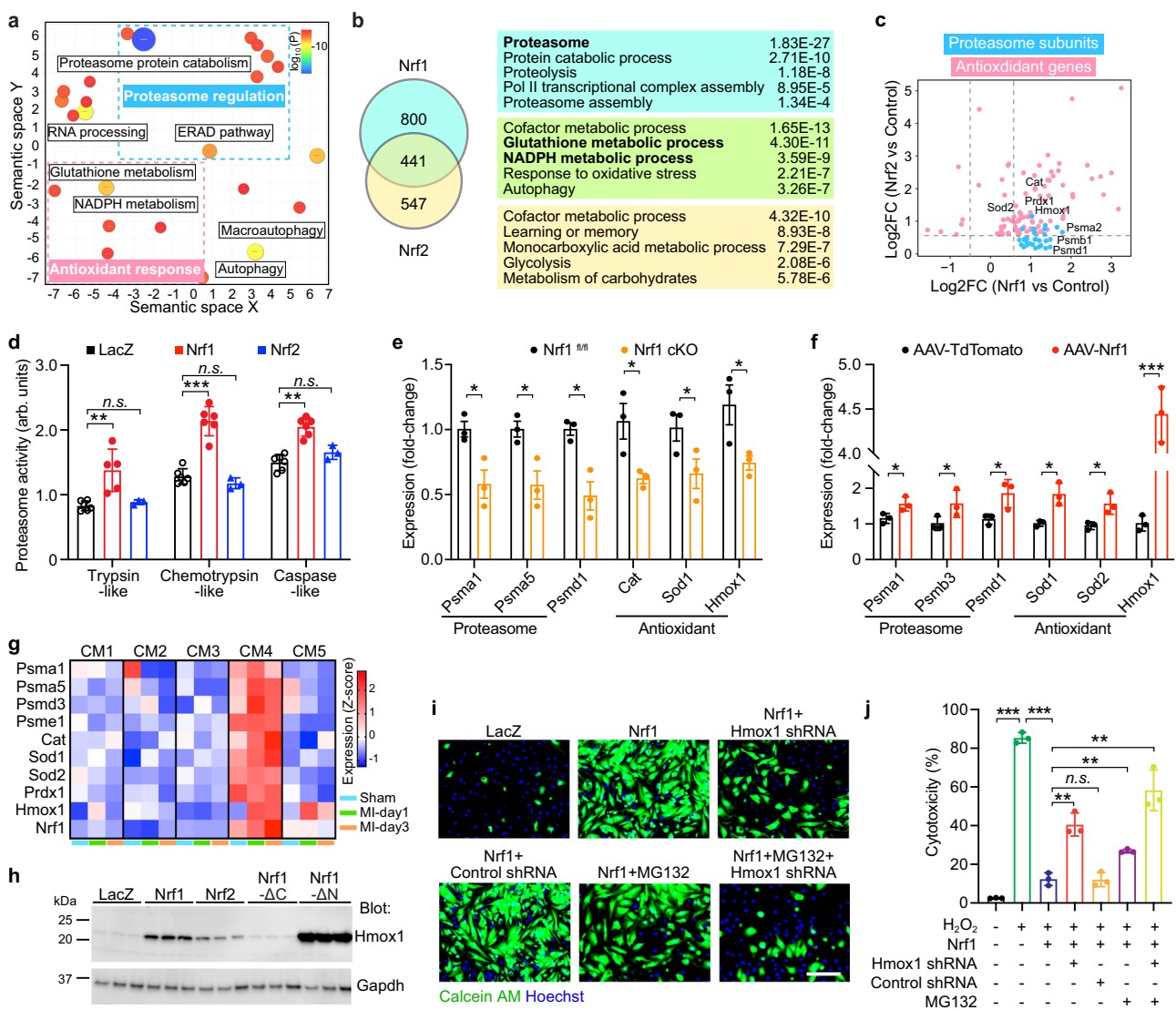

**Fig. 4 Nrf1 confers protection by regulating the proteasome and antioxidant response. a** Gene ontology analysis of genes upregulated in Nrf1 samples compared to LacZ control samples. Upregulated genes are displayed in semantic space as clusters; the color and proportional size of circles indicate log10 (P value). P values were obtained using Metascape (see the "Methods" section). **b** Left: Venn diagram showing a number of genes upregulated by Nrf1, Nrf2, or both; Right: top enriched GO terms for each gene group. **c** Log2 fold-change (FC) of selected proteasome subunit genes (blue) and antioxidant genes (pink) in NRVMs overexpressing Nrf1 (x-axis) or Nrf2 (y-axis). **d** Proteasomal activity in arbitrary units (arb. units) of NRVMs overexpressing LacZ, Nrf1 or Nrf2; n = 6 biologically independent experiments for groups LacZ and Nrf1; n = 3 biologically independent experiments for group Nrf2; Trypsin-like, **p = 0.0030; Chemotrypsin-like, ***p = 0.000009; Caspase-like, **p = 0.000031. **e** qPCR measurement of proteasome subunit genes and antioxidant genes in Nrf1 cKO and control hearts at P14; n = 3 for each group. Psma1, *p = 0.0266; Psma5, *p = 0.0261; Psmd1, *p = 0.0120; Cat, *p = 0.0364; Sod1,*p = 0.0462, Hmox1, *p = 0.0510. **f** qPCR measurement of proteasome subunit genes and antioxidant genes in AAV-Nrf1 and AAV-TdTomato hearts from 2-month-old mice; n = 3 for each group. Psma1, *p = 0.0459; Psmb3, *p = 0.0426; Psmd1, *p = 0.0507; Sod1, *p = 0.0128; Sod2, *p = 0.0268, Hmox1, ***p = 0.0001. **g** Expression of proteasome subunit genes and antioxidant genes in CM1–CM5 cells from our previous snRNA-seq data (GSE130699). Expression levels in z-scores from 1-day post-Sham (Sham), 1-day post-MI (MI-day1), and 3-day post-MI (MI-day3) samples were plotted. **h** Western blot showing Hmox1 expression in NRVMs overexpressing LacZ, Nrf1, Nrf2, Nrf1-ΔN, or Nrf1-ΔC. **i**, **j** Viable cells (**i**, green) and percent of cell death (**j**) in NRVMs overexpressing Nrf1 plus Hmox1 shRNA, or MG132 treatment, or both, following H₂O₂ treatment; n = 3 biologically independent experiments. ***p < 0.0001, **p = 0.002 (column 3 vs. 4 and column 3 vs. 7), **p = 0.0018 (column 3 vs. 6). Scale bar, 100 μm; **d**, **e**, **f**, **j** results are shown as mean ± s.e.m.; Student's t-test two-tailed; n.s. not significant.

redundant functions between Nrf1 and Nrf2, we generated mice with cardiomyocyte-specific knockout of Nrf2 (Nrf2 cKO) or both genes (dKO) and compared them with Nrf1 cKO mice. The expression of proteasome and antioxidant target genes was reduced to greater degrees in Nrf1 cKO compared to Nrf2 cKO hearts, and, importantly, was not further reduced in dKO hearts, suggesting that Nrf1 plays a dominant role in regulating the expression of these genes in the heart (Supplementary Fig. 12).

We further checked their expression in the five cardiomyocyte populations (CM1–CM5). Like Nrf1, these target genes were highly expressed in CM4 cells (Fig. 4g), suggesting that the proteasome and antioxidant response pathways are more active in CM4 cells than in other cardiomyocytes. Moreover, consistent with the transcriptional upregulation of these genes, proteomics analysis on NRVMs overexpressing Nrf1 revealed increased levels of proteins involved in the proteasome complex pathway and the

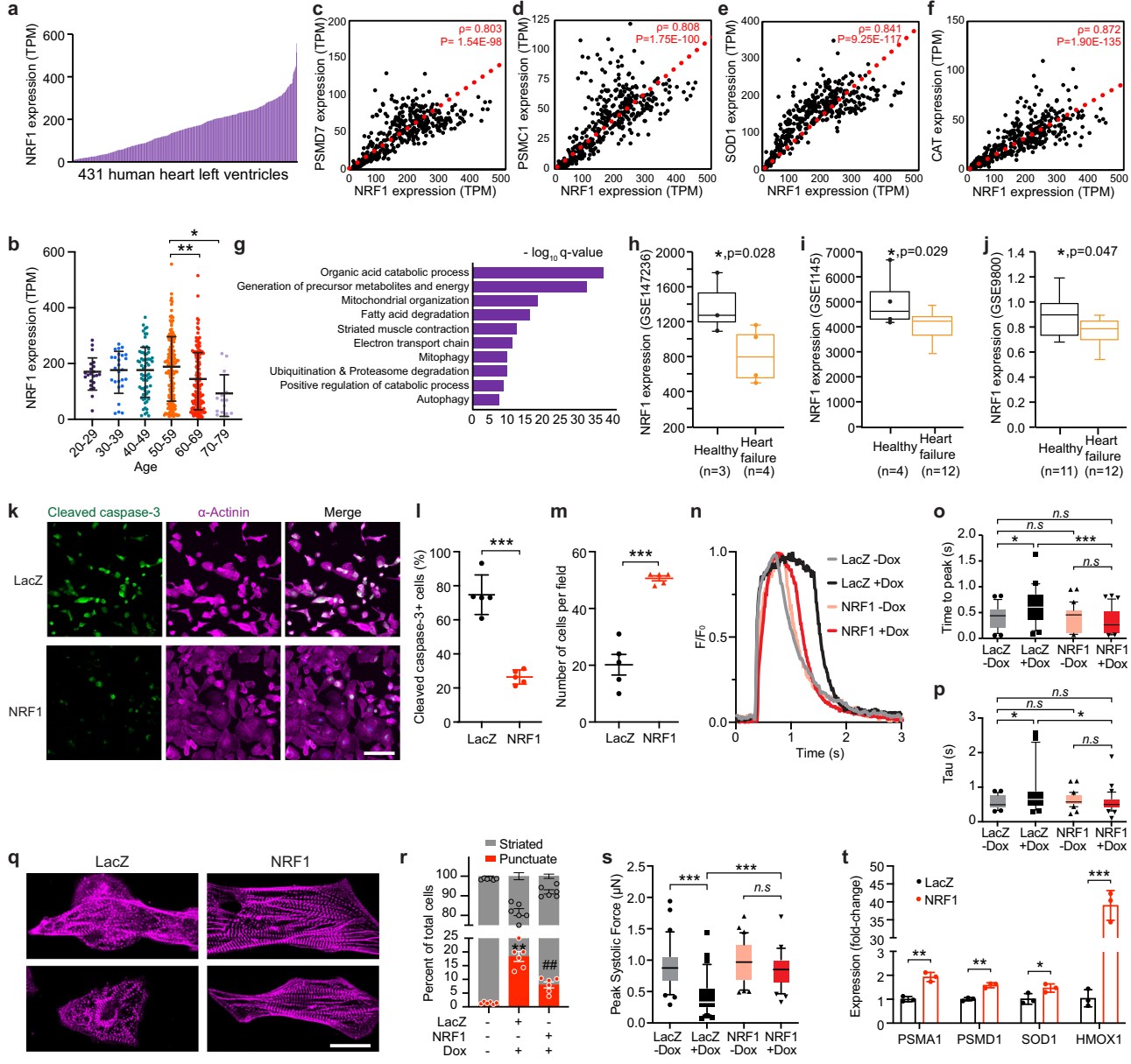

antioxidant response (Supplementary Fig. 13 and Supplementary Data 3).

Hmox1, a detoxifying enzyme that catabolizes heme to $Fe^{2+}$, CO, and biliverdin[26], was the most upregulated protein by Nrf1. Western blot analysis showed that Hmox1 expression positively correlated with the survival of cells overexpressing Nrf1 variants (Nrf1-ΔC and Nrf1-ΔN) or Nrf2 after $H_2O_2$ treatment (Fig. 4h). To study whether the expression of Hmox1 is required for cardioprotection by Nrf1, we co-expressed Nrf1 with an shRNA targeting *Hmox1* or a scrambled shRNA control in NRVMs followed by $H_2O_2$ treatment. The *Hmox1* shRNA reduced *Hmox1* transcripts by >80% (Supplementary Fig. 14a) and significantly attenuated the protective effective of Nrf1 (Fig. 4i, j).

To determine whether increased proteasomal activity also contributes to Nrf1 function, we incubated Nrf1-overexpressing NRVMs with a proteasome inhibitor, MG132[27], during $H_2O_2$ treatment. We observed a significant reduction of cell survival in MG132-treated cells (Fig. 4i, j, and Supplementary Fig. 14b, c). Cell survival was further reduced with combined *Hmox1* shRNA

and MG132 treatments, showing that both Hmox1 and proteasome activation mediate the protective function of Nrf1.

**Nrf1 protects hPSC-CMs from the cardiotoxin Dox**. Analysis of human heart ventricles from 431 donors (age 20–79) in the Genotype Tissue Expression (GTEx) database revealed varying levels (differing by over 70-fold) of *NRF1* expression among individuals (Fig. 5a), with a significant decline in the elderly demographic (age 60–79) (Fig. 5b). Gene co-expression analysis on these samples identified 1167 genes, whose expression highly correlated with *NRF1* (Spearman correlation coefficient > 0.8, adjusted *p*-val < 0.00001) (Supplementary Data 4). The correlated genes include *PSMC1*, *PSMD7*, *SOD1*, and *CAT*, which are known NRF1 targets (Fig. 5c–f). GO analysis revealed that these *NRF1* co-expressed genes are associated with key metabolic and energy production processes in the heart, such as fatty acid degradation, electron transport chain, and mitochondrial organization, as well as muscle contraction and known NRF1 downstream pathways—

**Fig. 5 NRF1 is an essential stress regulator in human iPSC-derived cardiomyocytes. a** Expression (TPM, transcript per million reads) of *NRF1* in 431 human heart left ventricle samples from the GTEx database. Samples were collected from 293 males and 138 females. **b** Expression of *NRF1* grouped by age. Age 20–29, $n = 22$; Age 30–39, $n = 26$; Age 40–49, $n = 66$; Age 50–59, $n = 154$; Age 60–69, $n = 149$; Age 70–79, $n = 14$; *$p = 0.0106$, **$p = 0.0027$, one-way ANOVA test. **c–f** Expression correlation plots showing that *PSMD7* (**c**), *PSMC1* (**d**), *SOD1* (**e**), and *CAT* (**f**) are highly co-expressed with *NRF1* in the 431 human heart left ventricle samples from the GTEx database. Spearman's correlation coefficient ($\rho$) and *p*-values are depicted; one-tailed *t*-test. **g** Top enriched GO terms for the 1167 *NRF1* co-expressed genes. **h–j** *NRF1* expression is significantly reduced in heart ventricular tissue from patients with heart failure compared to healthy hearts. Box-and-whisker plots are provided in which the central line denotes the median value, the edges of the boxes represent the upper and lower quartiles, and whiskers indicate the minimum and maximum values; *$p < 0.05$ with exact *p*-value depicted. **h** RNAseq data from GSE147236; $n = 4$ for heart failure, $n = 3$ for healthy control; two-tailed Wald test. **i** Affymetrix microarray data from GSE1145; $n = 12$ for heart failure, $n = 4$ for healthy control; Wilcoxon rank sum test two-tailed. **j** Agilent microarray data from GSE9800; $n = 12$ for heart failure, $n = 11$ for healthy control; Wilcoxon rank-sum test two-tailed. **k** Immunostaining of cleaved caspase-3 and α-actinin in hPSC-CMs overexpressing NRF1 or LacZ after Dox treatment. Scale bar, 100 µm. **l, m** Quantification of the percentage of cleaved caspase-3$^+$ cardiomyocytes (**l**) and total cell number (**m**) in hPSC-CMs overexpressing NRF1 or LacZ after Dox treatment. ***$p < 0.0001$, Student's *t*-test two-tailed. **n** Calcium transient in hPSC-CMs overexpressing NRF1 or LacZ with and without Dox treatment; $n = 29$–33 cells from two independent experiments were measured for each group. **o, p** Quantification of time-to-peak (**o**) and average decay time (Tau) (**p**) of calcium transient in hPSC-CMs overexpressing NRF1 or LacZ with and without Dox treatment; $n = 29$–33 cells from two independent experiments were measured for each group. **o** *$p = 0.0163$, ***$p = 0.0004$; **p** LacZ (−Dox) vs. LacZ (+Dox), *$p = 0.0479$; LacZ (+Dox) vs. Nrf1 (+Dox), *$p = 0.0416$; one-way ANOVA. **q** Immunostaining with α-actinin antibody showing sarcomere disarray in LacZ control cells and preserved sarcomere organization in NRF1-expressing cells after Dox treatment. Scale bar, 20 µm; **r** Quantification of the percentage of sarcomere disarray in hPSC-CMs overexpressing NRF1 or lacZ treated with Dox; $n = 481$ cells from six independent experiments were measured for the '−Dox' group; $n = 83$ cells from six independent experiments were measured for the 'LacZ + Dox' group; $n = 233$ cells from six independent experiments were measured for the 'NRF1 + Dox' group. **$p < 0.0001$ compared to untreated '−Dox' sample; ##$p = 0.001$ compared to 'LacZ + Dox' sample; Student's *t*-test two-tailed. **s** Peak systolic force of single hPSC-CMs overexpressing NRF1 or LacZ with (+Dox) and without (−Dox) Dox treatment; $n = 34$ for LacZ (−Dox), $n = 32$ for LacZ (+Dox), $n = 36$ for NRF1 (−Dox), and $n = 35$ for NRF1 (+Dox) from two independent experiments; ***$p < 0.0001$, one-way ANOVA. **t** qPCR measurement of proteasome subunit genes, *PSMA1* and *PSMD1*, and antioxidant genes, *SOD1* and *HMOX1*, in hPSC-CMs overexpressing NRF1 or LacZ; $n = 3$ for each group; *PSMA1*, **$p = 0.0019$; *PSMD1*, **$p = 0.0013$; *SOD1*, *$p = 0.0486$; *HMOX1*, ***$p < 0.0001$; Student's *t*-test two-tailed. **b, l, t** results are shown as mean ± s.d.; **m, r** results are shown as mean ± s.e.m.; **o, p, s** Box-and-whisker plots are provided in which the central line denotes the median value, the edges of the boxes represent the upper and lower quartiles, and whiskers indicate the 10–90 percentile values with points below or above the whiskers shown as individual dots. n.s. not significant.

ubiquitination, proteasome degradation, and autophagy (Fig. 5g). These data suggest that the degree of *NRF1* expression correlates with the metabolic and contractile properties of the heart, and thus likely affects cardiac function. Indeed, analysis of published datasets revealed a significant reduction of *NRF1* transcript levels in heart ventricles from three independent patient cohorts with heart failure ($n = 28$) (Fig. 5h–j).

Given that Dox-induced cardiomyopathy in cancer patients carries poor prognosis and is frequently fatal, we next studied whether NRF1 overexpression confers cardioprotection against Dox in hPSC-CMs. After treatment of hPSC-CMs with Dox, we observed significant cell apoptosis marked by cleaved caspase-3 and reduced cell numbers in control cells overexpressing LacZ (Fig. 5k–m). In contrast, NRF1 overexpression in hPSC-CMs significantly attenuated Dox-induced cell death, as seen by reduced activation of cleaved caspase-3 and improved cell viability (Fig. 5k–m and Supplementary Fig. 15). Consistent with the improved cardiomyocyte survival, calcium transients were markedly improved in NRF1-overexpressing cells, as both the rise-time-to-peak and average decay time measurements were significantly reduced compared to control cells treated with Dox, and were comparable to untreated cells (Fig. 5n–p). Additionally, NRF1-expressing cells exhibited less sarcomere disarray after Dox treatment (Fig. 5q, r), indicating preserved contractility. Indeed, the contractile force after Dox-treatment was significantly improved in hPSC-CMs overexpressing NRF1 compared to control cells (Fig. 5s). NRF1 overexpression increased the expression of *PSMA1*, *PSMD1*, *SOD1*, and *HMOX1* in hPSC-CMs (Fig. 5t), indicating a similar stress response mechanism that mediates the protective function of NRF1 in human cardiomyocytes.

## Discussion

A universal characteristic of dysfunctional hearts is elevated ROS and perturbed ER homeostasis[28,29]. Here, we demonstrate that

Nrf1 gain-of-function is sufficient to protect adult mouse hearts from I/R injury by activating ROS scavengers and increasing proteasomal activity to maintain oxidative and proteolytic stress balance. Due to its ability to activate a dual functional stress response, Nrf1 confers a greater degree of protection than Nrf2, a "master regulator" of redox balance, and thus is a core component of stress adaptation in the heart.

Although antioxidant response and proteasome activation are the two major pathways upregulated by Nrf1, the protection mediated by Nrf1 likely involves additional mechanisms. For example, our RNA-sequencing analysis also identified genes involved in autophagy, such as *Atg2a*, *Atg4d*, and *Atg13* that were upregulated by Nrf1 overexpression. Activation of autophagy has been shown to be cardioprotective[30,31], thus likely also contributing to the Nrf1-mediated protection. Recent studies suggested that Nrf1 also plays a role in maintaining metabolic homeostasis. In hepatocytes, Nrf1 was shown to directly regulate the expression of *Lipin1* and *PGC-1*, thus regulating lipid metabolism[32]. Additionally, NRF1 transgenic overexpression induced insulin resistance[33]. Whether similar functions of Nrf1 in regulating metabolism exist in the heart and contribute to its protective role warrants additional investigation. Furthermore, Nrf1 could also confer protection through anti-inflammation and anti-fibrosis pathways, as genes involved in innate immune responses and extracellular matrix organization were significantly downregulated by Nrf1.

We identified Nrf1 based on its enriched expression in CM4 cells. We have previously shown that CM4 cardiomyocytes are immature, enriched in the regenerative mouse heart, and enter the cell cycle in response to injury. In this study, we used spatial transcriptomic profiling, for the first time, to map the anatomic localization of cardiac cell types in mouse hearts undergoing regeneration. We showed that CM4 cells, compared to other cardiomyocyte populations, are proximal to the infarct site, suggesting their role in reconstituting the damaged myocardium.

Together, these results provide strong evidence that CM4 cells are the regenerative cardiomyocytes in neonatal mouse hearts, and the loss of CM4 cells in later life contributes to the loss of regenerative potential.

It is worth noting that several studies including ones from our lab have shown that BrdU$^+$ cardiomyocytes in regenerating hearts are localized throughout the ventricles[3,4,19]. However, to our knowledge, none of these studies provided direct evidence, such as lineage tracing, showing that BrdU$^+$ cardiomyocytes are in fact regenerating cardiomyocytes that ultimately reconstitute the damaged myocardium. The ubiquitous distribution of BrdU$^+$ cardiomyocytes could be attributed to the global binucleation and DNA synthesis events in cardiomyocytes during the neonatal stage, which cannot be easily distinguished from true proliferating cardiomyocytes that are induced by injury. In fact, when more faithful markers of cell proliferation, such as pH3, Aurora B, and clonal expansion were analyzed[3,34], we and others showed that proliferating cells are more frequently observed in the apex and border zones compared to remote zones, consistent with our results from the spatial transcriptome analysis. Additionally, in our previous paper[5], besides CM4, we also identified another cardiomyocyte population, CM2, that expresses high levels of cell-cycle genes. Although CM2 cells are detected at a low frequency (~3%), it is possible that they are localized more broadly, accounting for the observed BrdU$^+$ cardiomyocytes in remote zones. Additionally, one limitation of snRNA-seq is its inability to distinguish mono-nucleated versus bi-nucleated cardiomyocytes. This inherent technical limitation potentially affects our estimation of percentages of different cardiomyocyte populations especially in later postnatal stages such as P8, when the majority of cardiomyocytes become binucleated. Future experiments using large particle fluorescence-activated cell sorting will reveal any potential differences in the degree of their nucleation[35]. Nevertheless, the majority of cardiomyocyte binucleation arises from incomplete cell division instead of cell fusion, thus the two nuclei from the same cardiomyocyte should in principle share the same gene expression profile, which makes it unlikely to affect the transcriptomic analyses performed in this study.

In contrast to MI hearts, CM4 cells in Sham hearts were not anatomically confined and, instead, distributed sporadically in the ventricles. We cannot fully exclude the possibility that CM4 cells represent a transient transcriptome state that was activated in cardiomyocytes proximal to the infarct. However, given that CM4 cells are also present in Sham hearts, it is plausible that they migrate to the infarct site in response to injury. Indeed, several studies have shown that cardiomyocyte migration is required for heart regeneration[36–38]. Future studies to fate map CM4 cells during heart regeneration may provide insight into how regenerating cardiomyocytes might be recruited to repair the infarct tissue and what chemo-attractants might mediate this recruitment.

The necessity of Nrf1 for neonatal heart regeneration indicates an intricate interplay between cytoprotection and cardiac regeneration programs. Indeed, several signaling pathways, including Yap and PI3K–AKT pathways, were shown to promote both proliferation and survival of cardiomyocytes[39–42]. Yet, different from Yap and PI3K–AKT pathways, Nrf1 is not sufficient to promote cardiomyocyte proliferation by itself but rather provides a permissive state for cardiomyocytes to proliferate during regeneration. This is supported by two observations from our study: first, Nrf1 overexpression did not increase proliferation in cardiomyocytes, and second, Nrf1 cKO reduced the number of proliferating cardiomyocytes and impaired regeneration in neonatal hearts. Future rescue experiments by overexpressing Nrf1 and its target genes in Nrf1 cKO hearts will establish the role of Nrf1 in maintaining this regeneration permissive condition and further delineate the functional downstream pathways.

Together, our study uncovers a unique adaptive mechanism activated in response to injury that maintains the tissue homeostatic balance required for heart regeneration. Reactivating these mechanisms in the adult heart represents a potential therapeutic approach for cardiac repair.

## Methods

**Mice.** All mouse experiments complied with all relevant ethical regulations and were performed according to protocols approved by the Institutional Animal Care and Use Committees at the University of Texas Southwestern Medical Center (protocol 2016-101833 and 2017-102269). UT Southwestern uses the "Guide for the Care and Use of Laboratory Animals" when establishing animal research standards. All mice used in this study were housed at the pathogen-free Animal Resource Center at the University of Texas Southwestern Medical Center. All animals were bred inside an SPF facility with 12 h light/dark cycles with a temperature of 18–24 °C and humidity of 35–60% and monitored daily with no health problems. All animals were housed in groups of a maximum five per cage with ad libitum access to food and water. Nrf2$^{fl/fl}$ mice were obtained from the Jackson Laboratory (# 025433).

**Ethics statement.** The GTEx (https://gtexportal.org/home/datasets) v7 data were downloaded from the database of Genotypes and Phenotypes (dbGaP) of the NIH (project id phs000424.v7.p2). The access to the data followed the guidelines in the Data Use Certification Agreement.

**Neonatal MI.** All animal work described in this manuscript has been approved and conducted under the oversight of the UT Southwestern Institutional Animal Care and Use Committee. Timed-pregnant Nrf1$^{fl/fl}$:aMHC-Cre and Nrf1$^{fl/fl}$ mice in the C57BL/6 background were used to deliver pups for neonatal MI surgery at P3. Neonatal mice were anesthetized by hypothermia, and neonatal MI was performed as described previously[3,4,43]. Briefly, neonatal mice were anesthetized by hypothermia on an ice bed. Lateral thoracotomy at the fourth intercostal space was performed by blunt dissection of the intercostal muscles after skin incision. A tapered needle attached to a non-absorbable 8-0 suture (PROLENE, Ethicon) was passed through the mid-ventricle below the origin of the LAD coronary artery and tied to induce infarction. The pericardial membrane remained intact after LAD ligation. Myocardial ischemia is indicated by the light pallor of the myocardium below the ligature after suturing. After LAD ligation, neonates were removed from the ice, thoracic wall incisions were sutured with a 7-0 non-absorbable suture, and the skin wound closed by using skin adhesive. Sham-operated mice underwent the same procedure involving thoracotomy without LAD ligation.

**Adult I/R.** Timed-pregnant wildtype C57BL/6 mice were used to deliver pups for systemic AAV9 injection on P4 and I/R surgery at 8 weeks of age. AAV9 vectors expressing TdTomato or Nrf1 driven by the muscle-specific CK8 promoter were cloned by replacing the Cas9 coding sequence of the AAV9-CK8-Cas9 vector[20] with the corresponding coding sequences of TdTomato, or mouse Nrf1, and were packaged by the Harvard Medical School/Boston Children's Hospital Viral Core, as previously described[44,45]. C57BL/6 mice were injected with AAV9 at P4 at a titer of 5E13 vg/kg by intraperitoneal injection.

Heart ventricles from mice injected with AAV9-TdTomato or AAV9-Nrf1 were collected 4 weeks after injection to examine the overexpression levels of Nrf1. I/R surgery was performed on 8-week-old mice pre-injected with AAV9-TdTomato or AAV9-Nrf1. Mice were anesthetized with Ketamine/Xylazine complex, intubated, and ventilated with a MiniVent mouse ventilator (Hugo Sachs Elektronik; stroke volume, 250 μL; respiratory rate, 105 breaths per minute). Body temperature was carefully monitored with a rectal probe and maintained as close as possible to 37.0 °C. With the aid of a dissecting microscope, following left thoracotomy between the fourth and fifth ribs, a 7-0 nylon suture was passed under the left anterior descending coronary artery, and a nontraumatic occluder was applied on the artery. After 45 min of ischemia, the ligature was then released, the chest was closed in layers and the mice were allowed to recover.

**Transthoracic echocardiography.** Cardiac function was evaluated by two-dimensional transthoracic echocardiography on conscious mice using a VisualSonics Vevo2100 imaging system as described previously. FS and EF were used as indices of cardiac contractile function. M-mode tracings were used to measure LV internal diameter at end-diastole (LVIDd) and end-systole (LVIDs). FS was calculated according to the following formula: FS = [(LVIDd−LVIDs)/LVIDd]×100%. EF was calculated as EF = [(LVEDV−LVESV)/LVEDV]×100%. (LVESV, left ventricular end-systolic volume; LVEDV, left ventricular end-diastolic volume.) All measurements were performed by an experienced operator blinded to the study.

**TTC staining.** At the conclusion of reperfusion, the heart was excised and perfused with Krebs–Henseleit solution through an aortic cannula. To delineate infarcted from viable myocardium, the heart was then perfused with 1% TTC in phosphate buffer. To delineate the occluded/reperfused bed, the coronary artery was tied at

the site of the previous occlusion and the aortic root was perfused with 10% Phthalo blue dye. As a result of this procedure, the region at risk was identified by the absence of blue dye, whereas the rest of the left ventricle (LV) was stained dark blue. The LV was cut into five to six transverse slices, which were fixed in 10% neutral buffered formaldehyde, weighed, and photographed under a microscope. The corresponding areas were measured and from these measurements, infarct size was calculated as a percentage of the region at risk.

**Analysis of fibrotic tissue**. Hearts were fixed in 4% paraformaldehyde in PBS, cryopreserved in 30% sucrose/PBS, and embedded in O.C.T. Compound. Hearts were cross-sectioned at 10 μm intervals starting from the ligation suture. Sections were collected at 0, 200, 400, 600, and 800 μm below the suture were collected. Masson's Trichrome staining was performed on heart sections as described previously[11]. Afterimage acquisition of whole heart sections, the fibrotic region indicated in blue was quantified in Image J. Especially, the freehand selection tool was used to trace the fibrotic region, and the area was measured (Analysis → Measure). The total heart cross-sectional area was also traced using the freehand selection tool and then measured. The fibrotic area was calculated as the fibrotic region normalized by the heart section area.

**Immunohistochemistry**. Heart samples were fixed and sectioned as described above. Heart sections were air-dried for 30 min at room temperature, fixed with 4% PFA for 20 min, washed twice with PBS, and permeabilized with 0.3% Triton X-100 in PBS for 10 min. Sections were then blocked in 5% goat serum/3% BSA/0.025% Triton X-100/PBS blocking solution for 1 h and stained with the indicated primary antibodies prepared in blocking solution at 4 °C overnight using the following dilutions: cTnT (13-11) (Invitrogen, MA5-12960, 1:200), pH3 (Ser10) (Cell Signaling Technology, 9701S, 1:200), Nrf1 (Abcam, ab238154, 1:250), CD45 (30-F11) (Tonbo Biosciences, 70-0451, 1:250), Vimentin (280618) (R&D Systems, MAB2105, 1:250), cTnI (Abcam, ab47003, 1:200). Sections were subsequently washed with 0.025% Triton X-100/PBS three times and incubated with corresponding secondary antibodies: Goat anti-mouse IgG Alexa 647 (ThermoFisher Scientific A32728, 1:500), Goat anti-mouse IgG Alexa 488 (ThermoFisher Scientific A32723, 1:500), Goat anti-rabbit IgG Alexa 647 (ThermoFisher Scientific A32731, 1:500), Goat anti-rat IgG Alexa 488 (ThermoFisher Scientific A-11006, 1:500) and Goat anti-mouse IgG Alexa 647 (ThermoFisher Scientific A32733, 1:500), prepared in blocking solution at room temperature for 1.5 h. After secondary antibody incubation, sections were washed with PBS, incubated with DAPI at room temperature for 10 min, and washed twice with PBS before mounting. Images were obtained using a Zeiss LSM 800 confocal microscope. TUNEL assay was performed using Click-iT Plus TUNEL Assay for In Situ Apoptosis Detection kit (Thermo Fisher Scientific, C10619) following manufacturer's protocol.

**Generation of adenoviruses**. Adenovirus was generated using the Adeno-X Adenoviral System 3 (Clontech, 632267). Coding regions of β-galactosidase (lacZ), mouse Nrf1, human NRF1, and mouse Nrf2 were cloned into the pAdx-CMV vector containing a red fluorescent protein. Adenoviruses were generated by transfecting linearized recombinant adenoviral plasmids into a mammalian packaging cell line Adeno-X 293. Primary lysates were used to re-infect Adeno-X 293 cells to generate higher-titer viruses. The viral titer was determined in 292T cells using Adeno-X RapidTiter Kit (Clontech, 632250). Adenoviruses were stored in 20 μL aliquots at −80 °C.

**Cardiomyocyte in vitro survival assay**. NRVMs were isolated from 1- or 2-day-old Sprague-Dawley rats with the Isolation System for Neonatal Rat/Mouse Cardiomyocytes (Cellutron, nc-6031) according to the manufacturer's instructions. NRVMs were plated at a density of $3 \times 10^5$ cells/well to gelatin-coated 12-well plates and were maintained in DMEM/199 medium (3:1, 3% FBS) for 48 h before adenoviral infection. Adenoviruses were diluted in DMEM/199 medium containing 3% FBS and were added to NRVMs at an MOI of 50. Forty-eight hours after the infection, NRVMs were treated with 50 μM $H_2O_2$, 1 mM peroxynitrite (Cayman Chemical), 5 μM Dox (Sigma-Aldrich), or 10 μM erastin (Sigma-Aldrich) for 2, 2, and 24 h, respectively. At the end of the treatments, the culture medium from treated cells was transferred and the level of lactate dehydrogenase (LDH) released from damaged cells was quantified using the LDH Assay Kit (Abcam, ab65393). NRVMs were next washed with PBS and incubated with Calcein AM dye (Thermo Fisher, L3224) for 20 min following the manufacturer's protocol. Cell viability was then visualized on a Keyence BZ-X700 microscope.

To achieve Hmox1 knockdown in NRVMs, adenoviruses were generated to express shRNAs against rat Hmox1 under a U6 promoter. Three shRNAs were tested. shRNA #2 (target sequence 5-GATATCAGTGTGCAGAGATTT-3) showed the highest knockdown efficiency with ~80% reduction of Hmox1 expression and was chosen for the experiments shown in this study. NRVMs were co-infected with the Hmox1 shRNA adenoviruses and Nrf1 adenoviruses followed by $H_2O_2$ treatment and analysis of cell survival as described above.

To inhibit the proteasome, 10 μM MG132 was added to NRVM cultures. Prolonged inhibition of the proteasome can perturb proteostasis, which may sensitize cells to $H_2O_2$-induced toxicity or cause spontaneous cell death. Therefore, we only applied MG132 20 min before and during the 2 h $H_2O_2$ treatment, a condition that efficiently suppressed the proteolytic activity of the proteasome without compromising the viability of the cells without $H_2O_2$ treatment.

**Protein isolation and immunoblot analysis**. Dissected heart ventricles or cultured cells were homogenized on ice for 15 min in RIPA buffer followed by 20 min centrifugation at 12,000×*g* to pellet cell debris. Protein concentration in cell lysates was measured by BCA protein assay. 5–10 μg of protein was used for SDS–PAGE gel electrophoresis. After gel to membrane transfer, membranes were blocked in 5% milk for 30 min followed by primary antibody incubation at 4 °C overnight using the following dilutions: ubiquitin (P4D1) (Cell Signaling Technology, #3936, 1:5000), Hmox1 (Proteintech, 10701-1-AP, 1:500), Nrf1 (D5B10) (Cell Signaling Technology, #8052, 1:1000), and Nrf2 (Abcam, ab137550, 1:1000). After the primary antibody incubation, the protein-membrane was washed with 1× TBST buffer and incubated with the corresponding secondary antibody conjugated with horseradish peroxidase (HRP): Goat anti-mouse IgG HRP (abcam, ab6789, 1:1000) and Goat anti-rabbit IgG HRP (abcam, ab6721, 1:1000). Chemiluminescent signal was developed using SuperSignal™ West Femto Maximum Sensitivity Substrate.

**Analysis of oxidative stress**. To measure DNA damage, NRVMs were fixed with 4% PFA at room temperature for 15 min, permeabilized with 0.3% Triton X-100 in PBS for 20 min, followed by co-staining with antibody against 8-OHdG (Novus Biologicals, NB600-1508, 1:250) and antibody against cTnT (13-11) (Invitrogen, MA5-12960, 1:200) at 4 °C overnight. Donkey anti-mouse secondary antibody conjugated with Alexa 647 (Thermo Fisher Scientific, A32787, 1:500) and Donkey anti-goat secondary antibody conjugated with Alexa 488 (Thermo Fisher Scientific, A-11055, 1:500) were used. Images were acquired using a Zeiss LSM 800 confocal microscope.

To measure ROS levels in cells, live cells were incubated with DCFDA (Cellular ROS Assay kit, Abcam, ab113851) for 30 min and washed twice with the wash buffer. Fluorescent signals at excitation/emission 485/535 were recorded. To measure ROS levels in heart tissue lysate, the method using CM-H2DCFDA (Invitrogen) was used. Briefly, the homogenized lysate was prepared in 1× RIPA lysis buffer (DyneBio), supplemented with a protease inhibitor (cOmplete, mini, EDTA-free; Roche), and quantified for protein using a BCA protein assay kit (Pierce). Then, the lysate containing 100 μg of protein was incubated with 2 μM CM-H2DCFDA for 30 min at 37 °C in the dark. Fluorescence signals at excitation/emission 495/527 were recorded by a microplate reader. To measure mitochondrial ROS, live cells were incubated with 0.25 μM of mitochondrial ROS detection reagent from the Mitochondrial ROS Detection Assay Kit (Cayman Chemical) for 20 min at 37 °C. Cells were then treated with 50 μM $H_2O_2$ for 30 min at 37 °C in the dark. Fluorescence signals at excitation/emission 480/560 were recorded by a microplate reader.

**Analysis of proteasomal activity**. The Cell-Base Proteasome-Glo Assays (Promega) was used to measure proteasome activity in NRVMs according to the manufacturer's instructions. Chemotrypsin-like, trypsin-like, and caspase-like activity of the proteasome was measured independently using the corresponding cleavage substrates. The proteasome activity in tissue lysate was measured using the Proteasome Activity Assay kit (abcam) according to the manufacturer's instructions. Briefly, heart tissue was lysed in 0.5% NP-40 (without supplement of additional ATP). The tissue lysate was incubated with proteasome substrate (Succ-LLVY-AMC) at 37 °C for 30 min with or without MG132. The fluorescent signal was recorded to calculate the ATP-non-stimulable proteasomal activity, which is the MG132 inhibitable fraction of the signal.

**Tandem mass tag proteomics analysis**. Ten million NRVMs infected with Nrf1 or LacZ adenoviruses were collected for proteomic analysis. Samples were reduced in volume by half using a SpeedVac, then an equal volume of 10% SDS with 100 mM of triethylammonium bicarbonate (TEAB) buffer was added to each sample. Tris (2-carboxyethyl) phosphine (TCEP) was added to a final concentration of 10 mM and incubated at 56 °C for 30 min, followed by the addition of iodoacetamide to a final concentration of 20 mM, which was incubated at room temperature for 30 min in the dark. Solutions were then acidified with 12% phosphoric acid, followed by the addition of six times the volume of S-Trap binding buffer (90% methanol, 10% 1 M TEAB), were then loaded onto the S-Trap column (Protifi). The column was washed 3 times with 150 μl of binding buffer followed by the addition of trypsin (1:10) in 50 mM TEAB and overnight incubation at 37 °C. Peptides were recovered by eluting the s-Trap with 35 μl of 50 mM TEAB, 0.2% formic acid (FA), and 50% ACN in 0.2% FA, sequentially. The eluate containing the peptides was fractionated into 8 fractions using the Pierce High pH Reversed-Phase Peptide Fractionation Kit.

The resulting fractions were dried and then reconstituted in 2% (v/v) ACN and 0.1% trifluoroacetic acid in water. These were injected onto an Orbitrap Fusion Lumos mass spectrometer coupled to an Ultimate 3000 RSLC-Nano liquid chromatography system. Samples were injected onto a 75 μm i.d., 75-cm long EasySpray column (Thermo) and eluted with a gradient from 0% to 28% buffer B over 180 min. Buffer A contained 2% (v/v) ACN and 0.1% formic acid in water, and buffer B contained 80% (v/v) ACN, 10% (v/v) trifluoroethanol, and 0.1% formic acid in water. The mass spectrometer operated in positive ion mode with a

source voltage of 1.8 kV and an ion transfer tube temperature of 275 °C. MS scans were acquired at 120,000 resolution in the Orbitrap and the top 10 peaks with charge = 2–6 were selected for MS2 using collisionally induced dissociation with a collision energy of 35%. The top 10 fragments were selected for MS3 fragmentation using HCD (SPS-MS3), with a collision energy of 55%. Dynamic exclusion was set for 25 s after an ion was selected for fragmentation.

Raw MS data files were analyzed using Proteome Discoverer v2.4 (Thermo), with peptide identification performed using Sequest HT searching against the rat protein database from UniProt. Fragment and precursor tolerances of 10 ppm and 0.6 Da were specified, and three missed cleavages were allowed. Oxidation of Met was set as a variable modification and TMT6plex addition to K and peptide N-termini were set as static modifications. The reporter ion intensities for peptides identified for each protein were further analyzed using msms.edgeR to identified DEG with cutoffs fold-change >1.5 and FDR < 0.05.

**Human iPSC (hPSC) differentiation and Dox treatment.** hPSC derived from a healthy male donor was directly obtained from the UT Southwestern Wellstone Myoediting Core. hPSCs culture and differentiation were performed as previously described[46]. Briefly, hPSCs were cultured on Matrigel-coated tissue culture poly-styrene plates and maintained in Essential 8 media (Thermo Fisher). hPSCs were passaged at 70–80% confluency using Versene (Thermo Fisher). Cardiac differentiation of hPSCs was induced when hPSCs achieved confluency by treating cells with CHIR99021 (Selleckchem) in RPMI (Thermo Fisher) supplemented with B27 without insulin (Thermo Fisher) for 24 h (from day 0 to day 1). The medium was replaced with RPMI/B27-insulin on day 1. The cells were then treated with IWP4 (Stemgent) in RPMI/B27-insulin at day 3 and the medium was refreshed on day 5 with RPMI/B27-insulin. Cardiac myocytes were maintained in RPMI supplemented with B27 (Thermo Fisher) starting from day 7, with the medium changed every 2–3 days. Ascorbic acid at 50 μg/mL was added to both media. Metabolic selection of cardiac myocytes was performed for 6 days starting at day 10 of differentiation by culturing cells in RPMI without glucose (Thermo Fisher) supplemented with 5 mM sodium DL-lactate and CDM3 supplement[47].

hPSC-CMs were harvested on day 30 of differentiation by treating the cells with TrypLe Express (Thermo Fisher). Cells were plated onto 12-well plates at 300k cells per well or 35 mm glass-bottom dishes at single-cell density before infection with adenoviruses expressing human NRF1 and lacZ (control) on day 35 with MOI 25–50. Adenoviral infection efficiency was assessed 30 h after the infection and over 70% of cells were infected as indicated by a red fluorescent reporter. Forty-eight hours after the infection, hPSC-CMs were treated with 10 μM Dox.

**Calcium imaging and cell survival assessment in hPSC-CMs.** After Dox treatment for 1 h, cells were loaded with the fluorescent calcium indicator Fluo-4 AM (Thermo Fisher) at 2 μM. Spontaneous $Ca^{2+}$ transients of beating hPSC–CMs were imaged at 37 °C using a Nikon A1R + confocal system. $Ca^{2+}$ transients were processed using Fiji software and analyzed using Microsoft Excel and Clampfit 10.7 software (Axon Instrument). The calcium release phase was represented with time to peak, which was calculated as the time from baseline to maximal point of the transient. The calcium reuptake phase was represented with the time constant tau by fitting the decay phase of calcium transients with a first-order exponential function.

For analysis of cell survival, hPSC-CMs were treated with 10 μM Dox for 30 h. Calcein AM staining was done as described above. Additionally, cells were fixed in 4% PFA and immunocytochemistry staining was performed using antibodies against cleaved caspase-3 (Cell Signaling Technology, 9661, 1:250) and α-actinin (Sigma, A7811, 1:250).

**Contractile force measurements.** For single-cell peak systolic force measurements, hPSC-CMs were plated at single-cell density on flexible poly-dimethylsiloxane (PDMS) 527 substrates (Young's modulus = 5 kPa) prepared according to a previously established protocol[48]. Videos of contracting hPSC-CMs were captured at 37 °C using a Nikon A1R+ confocal system at 59 frames per second in resonance scanning mode.

Contractile force generation of single hPSC-CMs was quantified using a previously established method[49]. In brief, movies of single contracting hPSC-CMs were analyzed using Fiji. Maximum and minimum cell lengths during contractions as well as cell widths were measured, and peak systolic forces generated were calculated using a previously published customized Matlab code[50].

**Spatial transcriptome library preparation.** The spatial transcriptome analysis on heart sections was performed using the Visium Spatial Gene Expression Reagents Kits (10×Genomics) according to the manufacturer's protocol. Briefly, mouse hearts were dissected and immediately embedded in a tissue embedding mold filled with OCT and placed in a bath of isopentane and liquid nitrogen. OCT-embedded heart tissue blocks were cryosectioned in a cryostat to generate appropriately sized sections for Visium Spatial slides. 10 μm sections at 200 μm below the ligation suture in MI heart or corresponding positions in Sham hearts were collected and placed on the Visum Spatial slides capture areas. Additional 5–10 sections from the same heart were collected in a 1.5 mL Eppendorf tube for evaluation of RNA quality. After confirming the RNA quality using an Agilent 2100 TapeStation,

tissue sections on the Gene Expression Spatial slides were fixed in methanol and stained with Hematoxylin and Eosin. The stained tissue sections were imaged using Keyence BZ-X710. Heart sections were next permeabilized for 18 min followed by RNA reverse transcription. The optimized permeabilization time was determined using the Visum Spatial Tissue Optimization Kit (10xGenomics) according to the manufacturer's protocol. Second strand cDNA was subsequently synthesized on tissue sections and was transferred to an Eppendorf tube after denaturation. The cDNA from the tissue slide that was spatially barcoded was next amplified via PCR and enzymatically fragmented. P5, P7, i7, and i5 sample indexes are added via End Repair, A-tailing, Adaptor ligation, and PCR to make the final sequencing library. Sequencing was performed on an Illumina Nextseq 500 system operated by the Next Generation Sequencing Core of Children's Research Institute at UT Southwestern using the 150 bp high output sequencing kit (Illumina), with the following pair-end sequencing settings: Read1—28 cycles, i5 index—10 cycles, i7 index—10 cycles, Read2—90 cycles.

**Spatial transcriptome data analysis.** The Space Ranger Single-Cell Software Suit (https://support.10xgenomics.com/spatial-gene-expression/software/pipelines/latest/what-is-space-ranger) was used to perform sample demultiplexing, reads alignment, and to generate feature-spot matrices. The cDNA reads were aligned to the mm10/GRCm38 mRNA reference genome. H&E staining images were supplemented to Spatial Ranger for tissue detection, fiducial detection, and alignment of the spatial position of sequenced data points. Further analyses were performed using the Seurat R package v3.2, following the instruction (https://satijalab.org/seurat/v3.2/spatial_vignette.html). Specifically, data normalization was performed using sctransform, which builds regularized negative binomial models of gene expression to account for variance in sequencing depth across data points while preserving the biological spatial variance on tissue sections. Mapping statistics of each sample are shown in Supplementary Fig. 1a. To map the spatial location of a cell type that we previously identified in neonatal regenerating hearts (GSE130699)[5], we deconvoluted the spatial transcriptome by integrating it with the snRNA-seq gene expression matrix using the TransferData function in Seurat. This function implements an 'anchor'-based integration workflow that enables a probabilistic transfer of annotations from a reference (snRNA-seq data) to a query (spatial transcriptome) data set. This analysis generated probability scores for each cell type at each capture spot, which correlate with cellular fraction and were used to generate pie-charts to show cellular composition. The spatial locations of each cell type were visualized using SpatialFeaturePlot in Seurat.

**In situ hybridization using RNA-scope.** In situ hybridization was performed on frozen cryosections from mouse hearts at 3-day post P1 or P8 MI using the RNAscope Fluorescent Multiplex Kit (Advanced Cell Diagnostics). Briefly, fresh frozen tissue preparation and cryosectioning were performed as described above. Sections were fixed in 4% PFA for 15 min at 4 °C and subsequently dehydrated in 50%, 70%, and 100% ethanol. Sections were then pretreated with Protease IV for 30 min at RT and washed in PBS. Sections were next hybridized with a probe mixture containing Acta2-C1 and Tnnt2-C3 probes or containing Nrf1-C1 and Tnnt2-C3 probes for 2 h at 40 °C in a HybEZ Oven. Slides were rinsed in 1× Wash Buffer and sequentially incubated with AMP 1-FL, AMP 2-FL, AMP 3-FL, AMP 4-FL Alt A, and DAPI, according to the manufacturer's protocol. Samples were imaged using a ZEISS LSM800 microscope. Transcript levels of *Acta2* and *Nrf1* from 29–34 cardiomyocytes from three individual sections were quantified using Image J. Specifically, the ROI manager function was used to trace cells between fluorescent channels. The transcript levels were calculated as the mean intergraded density (signal normalized by the area) using the "measure" function in Fiji.

A 10 ZZ probe was generated to target the mouse *Nrf1* transcript at the region below:

CATACAATATGGCACCCAGTGCCCTTGACTCTGCTGATCTACCACCAC
CCAGCACCCTCAAGAAAGGTAGCAAGGAAAAGCAGGCTGACTTCCTGGA
CAAGCAGATGAGCCGAGATGAGCACAGAGCCCGAGCCATGAAGATCC-
CATTCACCAATGACAAGATCATCAACCTGCCTGTAGAAGAATTCAATGA
GCTGCTGTCCAAATACCAGCTGAGCGAGGCCCAGCTCAGCCTCATCCGG
GATATCCGGCGCCGGGGCAAAAACAAGATGGCTGCACAGAACTGCCGC
AAGCGCAAGTTGGACACCATCCTAAACCTAGAACGTGATGTGGAGGACT
TGCAGCGAGATAAGGCCCGATTGCTTCGAGAAAAGGTAGAGTTCCTTCG
GTCTCTGCGACAGATGAAGCAGAAGGTCCAAAGCTTATACCAGGAGGT
GTTTGGGCGGCTGGGGATGAGCATGGGAGGCCCTACTCACCCAGTC.
RNA-scope Probe-Mm-Tnnt2-C3 (418681-C3) and RNA-scope Probe-Mm-Acta2 (319531) were purchased from Advanced Cell Diagnostics.

**Cardiac nuclei isolation and single nucleus RNA-seq sequencing.** Tail genotyping was performed on the day of collection (P4, one day post-P3 surgery) to distinguish Nrf1^fl/fl and Nrf1^fl/fl:αMHC-Cre mice. Mice were then euthanized, and hearts were extracted. Each heart was dissected in ice-cold PBS and a transverse cut on the ligation plane (for MI hearts) or similar level (for sham hearts) was made. Heart tissue below the ligation plane was collected and pooled from 4 to 6 hearts of the same genotype for nuclei isolation. Cardiac nuclei isolation was performed as previously described[51]. Total cardiac nuclei were used to generate single nucleus

RNA-seq libraries using Single Cell 3′ Reagent Kits v3 (10×Genomics) according to the manufacturer's protocol.

**snRNA-seq data analysis**. The Cell Ranger Single-Cell Software Suit (https://support.10xgenomics.com/single-cell-geneexpression/software/pipelines/latest/what-is-cell-ranger) was used to perform sample demultiplexing, barcode processing, and single-cell 3′ gene counting. The cDNA reads were aligned to the mm10/GRCm38 pre-mRNA reference genome. We sequenced an average of 14,610 reads with over 90% mapped to the genome and a median of 1321 genes per nucleus (Supplementary Fig. 6a). Only confidently mapped reads with valid barcodes and unique molecular identifiers were used to generate the gene-barcode matrix. Further analyses for quality filtering were performed using the Seurat R package[52]. Doublets were identified using Scrublet and removed from downstream analysis[53]. The distribution of detected genes per cell in each sample is summarized in Supplementary Fig. 6b. For quantity filtering, we removed cells that had more than 2500 or fewer than 200 detected genes. We then normalized the data by the total expression, multiplied by a scale factor of 10,000, and log-transformed the result. To account for variation in the number of genes and UMI detected in each cell, we normalized the scaled expression matrix using the ScaleData function and set the vars.to.regress to nUMI and need. IntegrateData function implemented in Seurat V3 was used to correct batch effects between samples and to merge samples into one Seurat object. Briefly, we performed standard preprocessing (log-normalization) and identified the top 2000 variable features for each individual dataset. We then identified integration anchors using the FindIntegrationAnchors function. We used default parameters and dimension 30 to find anchors. We then passed these anchors to the IntegrateData function to generate an integrated Seurat object. To visualize the data, we used Uniform Manifold Approximation and Projection (UMAP) to project cells in 2D space on the basis of the aligned canonical correlation analysis[54]. Aligned canonical correlation vectors (1:30) were used to identify clusters using a shared nearest-neighborhood modularity optimization algorithm[52]. Cell clusters were identified using the FindCluster function in Seurat with resolution 0.3. Four cardiomyocyte clusters were identified based on the expression of Tnnt2 (Supplementary Fig. 6d) and were used for further analyses. In total, we analyzed 5176 cardiomyocyte nuclei from Nrf1 cKO hearts (MI + Sham) and 3409 cardiomyocyte nuclei from control hearts (MI + Sham).

To classify cardiomyocytes based on the CM1–CM5 populations that we previously identified in our snRNA-seq study, we followed the data transfer method developed in the Seurat package. The detailed procedure can be found at (https://satijalab.org/seurat/v3.2/integration.html). In brief, the FindTransferAnchors function was used to identify anchors between the query dataset (generated in this study) and the reference dataset (generated previously in ref. [5]). The identified anchors were used to transfer cell type labels learned from the reference data to the query data with the function TransferData. After label transfer, CM1 and CM3 cells were merged into one cluster (CM1/3) due to the lack of clear separation on the UMAP. The remaining clusters, CM2, CM4, and CM5, formed distinct clusters as visualized in the UMAP plot. The percentage of label transferred CM1/3, CM2, CM4 and CM5 cells in each sample were calculated and shown in Fig. 2i. Differential gene analysis was performed by comparing CM4 cells in Nrf1KO MI hearts to CM4 cells in the control MI heart ($p$-adjust < 0.1 and FC = 1.5) to identify genes that showed up-regulated and down-regulated expression in CM4 cells by Nrf1 deletion at 1-day post-MI (Supplementary Data 1). To calculate the correlation between Nrf1, Acta2, Mki67, and Ccnb2, sparse single nucleus expression data were imputed using Markov Affinity-based Graph Imputation of Cells (MAGIC https://github.com/KrishnaswamyLab/MAGIC) and robust spearman correlation method was used.

**Bulk RNA Sequencing in NRVMs and data analysis**. NRVMs were collected at 48 h after infection with adenoviruses for lacZ, Nrf1, and Nrf2. RNA was extracted using the RNeasy Mini Kit (QIAGEN, 74104) according to the manufacturer's protocol. An Agilent 2100 TapeStation was used for RNA quality analysis. Stranded mRNA-seq libraries were generated using the KAPA mRNA HyperPrep Kit (Roche, KK8581) following the manufacturer's protocol. Sequencing was performed on an Illumina NextSeq 500 system using a 75-bp, high-output sequencing kit for single-end sequencing.

For data analysis, quality control of RNA-seq data was performed using the FastQC Tool (Version 0.11.4). Sequencing reads were aligned to rat Rnor 6.0 reference genome using HiSAT2 (Version 2.0.4) with default settings and -rna-strandness F[55]. After removal of duplicate reads using Samtools v0.1.18[56], aligned reads were counted using featurecount (Version 1.6.0) per gene ID[57]. Differential gene expression analysis was performed with the R package edgeR (Version 3.20.5) using the GLM approach[58]. For each comparison, genes with more than 1 count per million (CPM) in at least three samples were considered as expressed and were used for calculating the normalization factor. Cutoff values of absolute fold-change >1.5 and false discovery rate <0.05 were used to select DEGs between sample group comparisons.

**Pathway and gene set enrichment analysis**. Enrichment analysis of gene sets was performed using the Metascape (https://metascape.org/) with the supply of upregulated or downregulated DEG. GO analysis results ($P < 0.05$) were visualized by using REVIGO (http://revigo.irb.hr/) in semantic space as cluster representatives, which stands for remaining terms after reduction of the redundancy. The interactive graph was also analyzed where highly similar GO terms were linked as representing the degree of similarity with width.

**Quantitative real-time PCR analysis**. Total RNA was extracted from NRVMs using Trizol and reverse transcribed using iScript Reverse Transcription Supermix (Bio-Rad) with random primers. The quantitative polymerase chain reactions (qPCR) were assembled using KAPA SYBR Fast qPCR Master Mix (KAPA, KK4605). Assays were performed using a 7900HT Fast Real-Time PCR machine (Applied Biosystems). Expression values were normalized to 18S mRNA and were represented as fold change. Oligonucleotide sequences of qPCR primers are listed in Supplementary Table 1.

**Reporting summary**. Further information on research design is available in the Nature Research Reporting Summary linked to this article.

## Data availability
All data presented in this study are available in the main text or the supplementary materials. Raw and analyzed RNA-sequencing data generated during this study are available in the Gene Expression Omnibus (GEO) repository (http://www.ncbi.nlm.nih.gov/geo/) and are accessible through GEO series accession number GSE163631. Raw proteomics data are available in MassIVE with accession number MSV000087183. Source data are provided with this paper. Previously published datasets used in this study include: GSE130699, GSE147236, GSE1145, GSE9800, GSE95755, and GTEx [https://gtexportal.org/home/]. Source data are provided with this paper.

## Code availability
The MATLAB code used to perform contractile force measurements of hPSC-CMs has been deposited to GitHub: https://github.com/DarisaLLC/Cardio.

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

## Acknowledgements

We thank Zhaoning Wang for contributing to the published snRNA-seq data generation; Dr. Gokhan S. Hotamisligil (Harvard University) for providing the Nrf1^fl/fl mice; Jose Cabrera for graphics; Drs. Jian Xu and Yoon Jung Kim from the Children's Research Institute at the University of Texas Southwestern Medical Center for performing the Illumina sequencing; Dr. Feng Wang for assistance analyzing the human patient datasets; Dr. Xiang Luo and Erica Niewold for help with isolating NRVMs; John Shelton from the Molecular Histopathology Core for help with histology; Dr. Andrew Lemoff from the Proteomics Core for help with the TMT proteomics analysis. The Genotype-Tissue Expression (GTEx) Project was supported by the Common Fund of the Office of the Director of the National Institutes of Health, and by NCI, NHGRI, NHLBI, NIDA, NIMH, and NINDS. The data used for the analyses described in this manuscript were obtained from dbGaP accession number phs000424.vN.pN. M.C. is supported by a K99/R00 pathway way to independence grant (NHLBI, K99HL153683). This work was supported by grants from the NIH (AR-067294, HL-130253, HL-138426, and HD-087351), the Foundation Leducq Transatlantic Networks of Excellence in Cardiovascular Research, and the Robert A. Welch Foundation (grant 1-0025 to E.N.O.).

## Author contributions

M.C., N.L., R.B.-D. and E.N.O. designed the experiments and overall study. M.C., N.L., R.B.-D. and E.N.O. wrote the manuscript. M.C., A.A., M.G.M. and W.T. performed the experiments. M.C. and K.C. performed the transcriptome and proteomics data analyses. M.C. performed the spatial transcriptome analysis. L.X. and X.X. analyzed the human patient sequencing data. All authors discussed the results and participated in the manuscript preparation and editing. Correspondence and requests for materials should be addressed to E.N.O. (eric.olson@utsouthwestern.edu).

## Competing interests

The authors declare no competing interests as defined by Nature Research.
