## [Peer Review File · Nature Communications]

REVIEWER COMMENTS

Reviewer #1 (Remarks to the Author):

Neonatal cardiomyocytes are able to adapt to injury and regenerate. However, the molecular mechanisms involved remain unknown. Cui et al. determined Nrf1 as one of the dominant factors regulating this process.

The experiments are well performed, the data are clearly presented and only minimal changes are required to publish the manuscript.

1. Is the CM4 a real representative of cardiomyocytes or rather myofibroblasts?
2. Nrf1 cKO hearts showed decreased cardiac function (Fig. 2...) is this due to the influence of Nrf1 on mitochondrial function? Is the improved fractional shortening and ejection fraction after I/R in AAV9-Nrf1 transfected animals due to improved mitochondrial function?
3. hPSC-CMs were treated with 10 uM doxorubicin – this is a very high concentration. Please comment.
4. The determination of Gata4 and alpha smooth muscle actin is missing . Not mentioned in the RT-PCR primer list.
5. The data in Extended Data Fig.2 are not clear. Please specify what you mean by: chymotrypsin-like proteasomal activity. (“0S or 26S proteasome, ATP-stimulable fraction, MG132 inhibitable fraction).
6. What do the authors mean by ‘increased ubiquitin aggregates’? Increased ubiquitinated proteins? (These are not necessarily aggregates.)

Reviewer #2 (Remarks to the Author):

This is an interesting manuscript reporting data from a spatial transcriptomic analysis performed in regenerating neonatal hearts aimed to identify the characteristics of replicating cardiomyocytes. These results significantly extend the information of a recently published paper by the same authors reporting the transcriptional signature of cells analysed by single-nucleus sequencing (DevCell 2020).

The manuscript is well written, pleasant to read and with plenty of information that can be useful for the broad number of laboratories interested in both cardiac regeneration and myocardial

protection. The manuscript is clearly technology-driven and most experiments are well performed and convincing.

My main perplexity, however, is pertinent to the content that the main message the manuscript wishes to convey. The fact that NRF1 is a master regulator of cell response to stress is well established and the results presented here reinforce this concept and provide convincing evidence that this is also true in cardiomyocytes. What I find less convincing, however, is the assumption that NRF1 might also be specifically involved in the regulation of cardiomyocyte proliferation and cardiac regeneration. I find quite obvious that the knock out of a factor that permits cells coping with stress might also impair cell replication, as this process itself generates formidable stress to the cells. The fact that replicating cardiomyocytes express this factor, therefore, is not surprising, as this is likely part of the whole set of factors that are required for proliferation to occur. This is also in line with what the authors reported themselves in the 2020 DevCell paper, namely that CM4 cells upregulate cell survival pathways following ischemic injury. Finding NRF1 in replicating cardiomyocytes, therefore, is an interesting correlation to report but does not imply a causative role for the factor in regeneration itself. Re-writing some part of the manuscript in light of these considerations would improve putting the manuscript results into a more objective perspective (starting from the title, as “mechanistic interplay” does not reflect what the manuscript really shows).

Specific points

I am surprised to find CM4 cells (replicating/regenerating cardiomyocytes) localised in the infarct area only. Data from different laboratories (including those from the same group) show that BrdU+, replicating cardiomyocytes stimulated by the infarct in neonates are present throughout the ventricle and are also present in atria. This discrepancy needs to be addressed.

What is the effect of NRF1 on mitochondrial function and the balance between glycolytic and fatty acid oxidation pathways? And through what genes does NRF1 impact on OXPHOS?

The difference in levels of NRF1 in CM1, CM4 and CM5 cardiomyocytes should be reported specifically in this manuscript.

What is the effect of the NRF1 knock-down on neonatal cardiomyocyte proliferation ex vivo, either basal or induced by pro-proliferative treatments (e.g. NRG1, microRNAs etc)?

The authors show that an AAV9-NRF1 vector protects from acute cardiomyocyte loss after MI. Multiple data indicate that cardioprotection is often conferred through the activation of protective autophagy. Does NRF1 specifically activate autophagy in cardiomyocytes?

Reviewer #3 (Remarks to the Author):

The present manuscript by Cui and colleagues follows up on a recent paper from the same authors (Cui et al, Developmental Cell 2020). A significant fraction of the initial analysis is based on the same snRNAseq dataset in neonatal mouse hearts, which is used this time to identify, within specific cardiomyocyte subpopulations, new targets involved in the regenerative potential of the heart. They identify a role for Nrf1 as a key player in the adaptative response to different stress conditions in cardiomyocytes.

Major points:

1. In Figure panel 1N: the authors should present the information for Nfr1 in a format similar to panel 1A? CM4 is a relatively small cardiomyocyte population and many cells are expressing Nfr1 at high levels. Do the seemingly 2 populations observed in this scatter plot correspond to CM2 and CM4?
2. In Figure 2A and subsequent experiments. What is the rationale for performing MI at P3, compared to P1 MI performed in previous figures?
3. Figure 2h-i. This reviewer understands that the number of mice on which Figure 2h-i is based was 4-6 per genotype (and experimental group; i.e. sham/MI?). Hearts are pooled for the analysis, leaving Figure 2i subject to the presence of outliers.

- Extended data Figure 5a, which complements data Fig 2h-i shows a low number of CMs in WT-sham (n=1,406) compared to cKO-sham (n=3,328). While percentage numbers seem similar, what is the potential effect of this for the detection of lower abundance CM populations? For example, CM2 accounts for perhaps 3% of CM, which is some 40 cells.

- The axis is split in two segments in this panel. Please split the bars accordingly to avoid misinterpretations. Regarding this panel, the authors write "after MI, the Nrf1 cKO hearts showed a significant increase of CM5 cells". This seems visually clear, but it is not clear how this significance is achieved on a pooled sample?

- Last, while Extended data Figure 5a-b shows a similar median number of genes per cell when comparing ShamWT vs shamKO (and in MI), the distribution shown in panel 5b displays obvious differences in top percentiles. How does this affect data quality and interpretation?

-

4. Better descriptions are needed for Supplementary Tables, including n-numbers.

Regarding Supp Table 1, which complements Fig2L, the number of genes and GO terms identified would benefit from a side comparison of CM4 cell populations before MI. A full table for the enriched GO terms presented in Fig2L would be required, and it should include fold-enrichments and number of genes affected (i.e. in background gene list and in altered genes).

5. While it is relevant for CM4, the increase in ROS levels shown in Fig2M cannot be attributed (or immediately linked) to the observations made in CM4 cells as this may be a general effect, specially considering that CM4 are a relatively minor CM population. Have the authors check this in other CM cells?

6. Data presented in Figure 3 is remarkable, and the positive effect of Nrf1 delivery seems clear. However, a few questions arise from this figure:

- Do adult LV cardiomyocytes split into different populations? Although they will be terminally differentiated, the benefits of Nrf1 delivery might be associated to particular adult CM populations. The authors do not perform additional snRNAseq in adults.

- AAV injection is provided at postnatal day P4, but I/R experiments are performed when mice are 8 weeks old. This is important. AAV vectors will have a great tropism for post-mitotic cells and, at P4, will not reach proliferative cardiomyocytes (i.e. CM4) with a great avidity. Given the time at which Nrf1 is provided, it is plausible that some CM populations with no proliferative capacity at the time of delivery will be contributing to the observed effect. It would be interesting to see which CM populations uptake the vector. For this, perhaps a few key markers for each population would suffice. This would better link the experiments performed in adults with those performed in neonates in Figure 2.

- As mice will have non-physiological levels of Nrf1 for 8 weeks until I/R is induced, can the authors elaborate on why they chose P4 as the best time for AAV delivery?

- Does AAV9 delivery improve the outcome in Nrf1-cKO mice?

7. Although not as pronounced, Nfr2 confers a remarkable level of protection as shown in Figure 3K. What is the role of Nfr2 or even Nfr3 (not expressed in P14 hearts) in Nrf1-cKOs? As the authors nicely show, Nrf1 has unique roles in regulating the proteasome. Do any of the Nfe2 family genes exhibit compensatory effects, at least in physiological conditions, for the roles on which they overlap (i.e. regulation of RedOx balance)?

8. Extended Data Figure 10. Why do the authors use log₂ peptide counts in a proteomics analysis involving TMT labelling? Looking at the methods provided, the addition of TMT does not seem obvious to me. I would expect a quantification based on reporter ion intensities (i.e. 6-plex TMT with two groups of 3 samples each). Please revise.

9. Calcium transient experiment in Figure 5n is based on n=29-33 cells per group. How many times was this experiment repeated? This also applies to panels 5o and 5p, based on the same experiment.

10. Figures 1F to 1M are based on single heart sections. Can the authors show additional images from different hearts to demonstrate reproducibility?

Minor.

- Can the authors comment on the impact of CM bi-nucleation in snRNAseq experiments? This is more relevant as CMs exit the earliest postnatal stage and a higher percentage become binucleated.

- Please spell out abbreviation of Nrf1 in the abstract.

Reviewer #4 (Remarks to the Author):

Cui et al in their ms: Mechanistic interplay between cardioprotection and heart regeneration mediated by Nrf1, explore this role of Nrf1 in infarcted neonatal mouse heart. A dual stress response mechanism is described involving activation of the proteasome and maintenance of redox balance. A mechanistic interplay between adaptive

stress responses and heart regeneration is described and highlight the central role of Nrf1 in these processes. This findings are truly original and are based on the combination of state of the art methods.

To describe the location of cellular changes the authors use snRNA characterized cell types from ref 7 figure S1 where is described marker genes from neonatal mouse heart for 5 types of cardiomyocytes, epicardial, mural, immune, endothelial, endothelial/endocardial and fibroblast cells.

Further the authors describe spatial transcriptomics analysis performed by standard procedures using the Visium Spatial Expression Reagents Kits including evaluation of RNA quality using an Agilent 2100 TapeStation.

Likewise spatial transcriptomics data analysis was done by standard procedures with the space Ranger Single-Cell Software Suit. Further analyses was done with the Seurat R package. The spatial transcriptome was deconvoluted by integrating with the snRNA-seq gene expression matrix using Seurat functions TransferData enabling predictions scores for each cell type for each data point. Spatial locations of each cell type were then visualized using Seurat SpatialFeaturePlot.

Figure 1 shows the results from the spatial transcriptomics analysis. It shows that CM4 is upregulated one day after LAD ligation and this is verified with Acta2 RNA-scope analysis. The figure shows nicely where the infarction scar area is located in the HE image of the heart. Spatial distribution of the projected snRNA-seq defined celltypes is shown individually and also as pie charts at every data point with the proportion of the different cell types.

Although the distributions look logical, no description is given of which snRNA-seq markers genes were used for the different cell types (Only generally as ref 7). Also, no validation is presented on the accuracy of the proportions of cell types in each data point. Typically in each data point there should be 10-15 cells. Given the size of cardiomyocytes it might be fewer cells. Another unclarity is the use of snRNA data, that is single nuclear data for the deconvolution of cell types. Cardiomyocytes typically are polynuclear, that infer that there are less cardiomyocytes than cardiomyocyte nuclei in the data points. One uncertainty in this respect is if the different cardiomyocyte types have different degrees of polynucleation. Further, it is questionable that some pie charts show only proportions of 5-10% for some cell types. Although the overall results seem feasible, the projection of snRNA seq cell type profiles is not trivial and would need a more precise and validated description.

Christer Sylvén

REVIEWER COMMENTS

We would like to express sincere gratitude for the time and thoughtfulness put forth to improve our work. We have addressed all of the reviewers' comments by performing several additional experiments and revising the text. We appreciate your evaluation and invitation to submit this response and revision. A detailed point-by-point response is below.

Reviewer #1 (Remarks to the Author):

Neonatal cardiomyocytes are able to adapt to injury and regenerate. However, the molecular mechanisms involved remain unknown. Cui et al. determined Nrf1 as one of the dominant factors regulating this process. The experiments are well performed, the data are clearly presented and only minimal changes are required to publish the manuscript.

Response: Thanks for your support and enthusiasm!

1. Is the CM4 a real representative of cardiomyocytes or rather myofibroblasts?

Response: Our results show that CM4 cardiomyocytes are bona fide cardiomyocytes in neonatal hearts and are not myofibroblasts. First, CM4 cells are present in both MI and Sham hearts (Fig. 2i), whereas myofibroblasts are activated in response to stimuli, such as injury, and thus are not usually present in uninjured hearts. Second, in our previous study (Cui et al. 2020), we showed that CM4 cells are enriched in P1 hearts and substantially decrease their abundance in P8 hearts. The disappearance of CM4 cells in P8 hearts is in contrast to the more pronounced activation of myofibroblasts in P8 hearts compared to P1¹. Lastly and most importantly, our gene expression analysis showed that CM4 cells express high levels of canonical cardiomyocyte marker genes, such as Tnnt2 (Supplementary Fig. 6d).

We agree with the reviewer and also think that it is interesting that CM4 cells express higher levels of ECM genes compared to other CM populations. The expression of ECM genes may suggest a role of CM4 cells in remodeling the extracellular environment during heart regeneration, consistent with several reports showing that the extracellular environment is critical to determining the ability of neonatal hearts to regenerate^{2,3}. It is plausible that regenerative CM4 cardiomyocytes contribute to the establishment and maintenance of this regeneration-prone extracellular environment, and we have included this discussion in our previous published study (Cui et al. 2020).

2. Nrf1 cKO hearts showed decreased cardiac function (Fig. 2...) is this due to the influence of Nrf1 on mitochondrial function? Is the improved fractional shortening and ejection fraction after I/R in AAV9-Nrf1 transfected animals due to improved mitochondrial function?

Response: In our snRNA-seq of the Nrf1 cKO hearts, we found that the OXPHOS pathways, including ATP metabolic process, respiratory electron transport, and the TCA cycle were upregulated in CM4 cells of Nrf1 cKO hearts compared to control hearts at 1-day after injury (Fig. 2l). The upregulation of OXPHOS pathways indicates a shift of mitochondrial energy production to an aerobic state that generates ROS, which is known to inhibit cardiomyocyte proliferation and heart regeneration. Consistently, we observed increased ROS levels and impaired heart regeneration in Nrf1 cKO hearts after injury, as shown in Fig. 2 m and d.

Although we did not directly measure mitochondrial function in the AAV9-Nrf1 hearts after I/R, given that Nrf1 increases expression of many antioxidant enzymes, it likely counteracts ROS, therefore protecting mitochondrial functions. To directly test if Nrf1 could alleviate mitochondrial ROS levels, we treated NRVMs overexpressing Nrf1 or LacZ (control) with H₂O₂ followed by MitoSOX staining, which specifically detects mitochondrial ROS production. We found that Nrf1 overexpression significantly reduced levels of Mitochondrial ROS induced by H₂O₂. These results are now shown in the **revised Supplementary Fig. 10 d**. We have also provided a clearer description of the result and Methods on **pages 10 and 23-24**, respectively.

3. hPSC-CMs were treated with 10 uM doxorubicin – this is a very high concentration. Please comment.

Response: Thank you for bringing up this important point. The dosage used for doxorubicin treatment varies in the literature but is generally between 0.1uM to 10 uM. The lower the concentration, the longer it takes for doxorubicin to induce toxicity in hPSC-CMs⁴. With doxorubicin at 10 uM, we see cytotoxicity in hPSC-CMs as early as 24h after treatment. We analyzed cell survival at 30h, which is a time point when most control cells died. Our experimental scheme allowed us to more rapidly test the cytoprotective effect of Nrf1. In fact, because of the high concentration we used in our study, we are extremely confident about the protective role of Nrf1.

4. The determination of Gata4 and alpha smooth muscle actin is missing. Not mentioned in the RT-PCR primer list.

Response: We apologize if our results were not presented clearly. In our study, we did not assess Gata4 expression. We measured alpha smooth muscle actin (Acta2) using RNA-FISH (RNA-scope) and listed the probe sequence in the “In situ hybridization using RNA-scope” section of the Methods.

5. The data in Extended Data Fig.2 are not clear. Please specify what you mean by: chymotrypsin-like proteasomal activity. (“0S or 26S proteasome, ATP-stimulable fraction, MG132 inhibitable fraction).

Response: Thank you for this comment. We have modified the figure legend to specify that it was the chymotrypsin-like activity of the 20S proteasome. We have further included details about conditions of ATP stimulation and MG132 inhibition during the assay in the Methods (**page 24**) of our revised manuscript.

6. What do the authors mean by ‘increased ubiquitin aggregates’? Increased ubiquitinated proteins? (These are not necessarily aggregates.)

Response: Thank you for this comment. We have now changed ‘increased ubiquitin aggregates’ to ‘increase ubiquitinated proteins’ in our revised manuscript (**page 6**).

Reviewer #2 (Remarks to the Author):

This is an interesting manuscript reporting data from a spatial transcriptomic analysis performed in regenerating neonatal hearts aimed to identify the characteristics of replicating cardiomyocytes. These results significantly extend the information of a recently published paper

by the same authors reporting the transcriptional signature of cells analysed by single-nucleus sequencing (DevCell 2020).

The manuscript is well written, pleasant to read and with plenty of information that can be useful for the broad number of laboratories interested in both cardiac regeneration and myocardial protection. The manuscript is clearly technology-driven and most experiments are well performed and convincing.

Response: Thanks for your support and enthusiasm!

My main perplexity, however, is pertinent to the content that the main message the manuscript wishes to convey. The fact that NRF1 is a master regulator of cell response to stress is well established and the results presented here reinforce this concept and provide convincing evidence that this is also true in cardiomyocytes. What I find less convincing, however, is the assumption that NRF1 might also be specifically involved in the regulation of cardiomyocyte proliferation and cardiac regeneration. I find quite obvious that the knockout of a factor that permits cells coping with stress might also impair cell replication, as this process itself generates formidable stress to the cells. The fact that replicating cardiomyocytes express this factor, therefore, is not surprising, as this is likely part of the whole set of factors that are required for proliferation to occur. This is also in line with what the authors reported themselves in the 2020 DevCell paper, namely that CM4 cells upregulate cell survival pathways following ischemic injury. Finding NRF1 in replicating cardiomyocytes, therefore, is an interesting correlation to report but does not imply a causative role for the factor in regeneration itself. Re-writing some part of the manuscript in light of these considerations would improve putting the manuscript results into a more objective perspective (starting from the title, as “mechanistic interplay” does not reflect what the manuscript really shows).

Response: Thank you for your critical suggestion. We have extensively re-written our manuscript (pages 3-4, 14-16), including a much more extended Discussion, to better put our discoveries into a more objective perspective. We have also changed the title of our manuscript to “Nrf1 promotes heart regeneration and repair by regulating proteostasis and redox balance”.

Specific points

1. I am surprised to find CM4 cells (replicating/regenerating cardiomyocytes) localised in the infarct area only. Data from different laboratories (including those from the same group) show that BrdU⁺, replicating cardiomyocytes stimulated by the infarct in neonates are present throughout the ventricle and are also present in atria. This discrepancy needs to be addressed.

Response: Thank you for bringing up this important point. We agree with you that several studies including the ones from our lab have shown that BrdU⁺ cardiomyocytes in regenerating hearts are localized throughout the ventricles. However, to our knowledge, none of the previous studies provided direct evidence, such as lineage tracing, to show that these BrdU⁺ cardiomyocytes are in fact regenerating cardiomyocytes that reconstitute the damaged myocardium. The ubiquitous distribution of BrdU⁺ cardiomyocytes is likely due to the global binucleation and DNA synthesis in cardiomyocytes during the neonatal stage which cannot be distinguished (using BrdU staining) from true proliferating cardiomyocytes that are induced by injury. In fact, when more faithful markers of cell proliferation, such as pH3, Aurora B, and

clonal expansion^{5,6}, were analyzed, we and others showed that proliferating cells are more frequently observed in the apex and border zones compared to remote zones. Additionally, in our previous paper (Cui et al 2020), besides the CM4 population, we also identified another cardiomyocyte population, CM2, that expresses high levels of cell-cycle genes. It is possible that CM2 cells are localized more broadly, accounting for the observed BrdU⁺ cardiomyocytes in the remote zone. We have included these points in **page 15 of the revised manuscript**.

To fully address this question, lineage tracing of proliferating cardiomyocytes during heart regeneration is required. Unfortunately, current lineage tracing models based on the expression of Mki67^{7,8} or Ccnb1⁹ are not all ideal to study cardiomyocyte proliferation due to their inability to distinguish cytokinesis from binucleation or polyploidization. Therefore, we plan to fate map CM4 cells by engineering a mouse model that has inducible Cre expressed under the control of the CM4 marker gene *Acta2*. This model will allow us to lineage trace CM4 cells and to study the distribution of their progenies during heart regeneration. Given the amount of work involved in these experiments, we will report these results in a separate study.

2. What is the effect of NRF1 on mitochondrial function and the balance between glycolytic and fatty acid oxidation pathways? And through what genes does NRF1 impact on OXPHOS?

Response: We analyzed the global effect of NRF1 overexpression in cultured cardiomyocytes using bulk RNA sequencing (Fig. 4 a and b) and TMT quantitative proteomics (Supplementary Fig. 12). Pathway analysis on differentially regulated genes/proteins showed that the most upregulated pathways are related to proteasome activation and antioxidant response. Pathways regulating glycolysis and fatty acid oxidation were not enriched for the NRF1-regulated genes/proteins. However, we cannot exclude the possibility that NRF1 may indirectly affect the activity of proteins involved in these metabolic pathways, such as regulating protein modifications. To definitively answer this question, metabolomics profiling would be required. Given the complexity of such metabolomics profiling experiments and the restricted time allowed for this revision, we decided not to include these experiments in our current study. However, we do want to point out that there are studies suggesting that NRF1 can regulate metabolic homeostasis. In hepatocytes, NRF1 was shown to directly regulate the expression *Lipin1* and *PGC-1*, thus regulating lipid metabolism¹⁰. Additionally, NRF1 transgenic overexpression induced insulin resistance¹¹. Whether similar functions of NRF1 in regulating metabolism exist in the heart requires further study. We have included these points in **page 14 of the revised manuscript**

3. The difference in levels of NRF1 in CM1, CM4 and CM5 cardiomyocytes should be reported specifically in this manuscript.

Response: Thank you for this suggestion. We have now included the expression of NRF1 in all CM populations (CM1-CM5) in **revised Supplementary Fig. 4c and Fig. 4g** of the revised manuscript.

4. What is the effect of the NRF1 knock-down on neonatal cardiomyocyte proliferation ex vivo, either basal or induced by pro-proliferative treatments (e.g. NRG1, microRNAs etc)?

Response: To answer your questions, we generated adenovirus expressing shRNAs targeting human NRF1, which effectively reduced NRF1 transcript levels in hPSC-CMs verified by qPCR (Rebuttal Figure a). We next studied the effect of NRF1 knock-down on cardiomyocyte

proliferation induced by nYap overexpression (Ad-nYap), a constitutive nuclear form of Yap that is pro-proliferative^{12, 13}, as well as at the basal level by LacZ (control) overexpression (Ad-LacZ). To assess proliferation, 48h after adenoviral infection, we pulse labeled cells with EdU for 24h. As shown below, nYap overexpression significantly increased the number of EdU⁺ CMs in both scrambled control and NRF1 shRNA treated cells, suggesting that NRF1 knockdown does not affect nYap-induced proliferation. However, compared to control shRNA, NRF1 shRNA reduced EdU⁺ CMs (p=0.013) in control cells overexpressing LacZ, suggesting that NRF1 knockdown reduced cardiomyocyte proliferation at the basal level. Reduced proliferation by the loss of NRF1 expression is consistent with NRF1 cKO hearts, which showed decreased numbers of pH3⁺ cardiomyocytes following injury (Fig. 2f). Given that this information is already presented in our *in vivo* studies and that nYAP is not a focus of our story, we would like to kindly request that the new experiments shown in the Rebuttal Figure are only submitted as part of our response to the reviewer and not included in the revised manuscript.

Rebuttal Figure. Effect of NRF1 knockdown on cardiomyocyte proliferation. **a**, qPCR measurement of NRF1 expression in hPSC-CMs infected with adenovirus expressing scrambled (control) or NRF1 shRNAs. **b**, EdU (violet) and cTnT (green) immunostaining of hPSC-CMs overexpressing LacZ or nYAP treated with control shRNA or NRF1 shRNAs. In this experiment, cells were first infected with adenovirus to express Lac or nYAP with control shRNA or NRF1 shRNA for 48h, followed by EdU incorporation for 24 h. **c**, Quantification of number of EdU⁺ cardiomyocytes in treatment conditions from **b**. **a**, **c**, results are shown as mean ± s.d; n.s., not significant; *p<0.05, ***p<0.001 by Student's t-test.

5. The authors show that an AAV9-NRF1 vector protects from acute cardiomyocyte loss after MI. Multiple data indicate that cardioprotection is often conferred through the activation of protective autophagy. Does NRF1 specifically activate autophagy in cardiomyocytes?

Response: Thank you for this question. Yes, in our RNA-seq analysis of NRF1-overexpressing cardiomyocytes, we did observe upregulation of autophagy pathways, which is shown in Fig. 4b. Specifically, we found that genes involved in autophagy, such as Atg2a, Atg4d, and Atg13, were significantly upregulated by NRF1 overexpression, indicating activated autophagy.

We have now included this discussion in our revised manuscript (page 14).

Reviewer #3 (Remarks to the Author):

The present manuscript by Cui and colleagues follows up on a recent paper from the same authors (Cui et al, Developmental Cell 2020). A significant fraction of the initial analysis is based on the same snRNAseq dataset in neonatal mouse hearts, which is used this time to identify, within specific cardiomyocyte subpopulations, new targets involved in the regenerative potential of the heart. They identify a role for Nrf1 as a key player in the adaptative response to different stress conditions in cardiomyocytes.

Major points:

1. In Figure panel 1N: the authors should present the information for Nrf1 in a format similar to panel 1A? CM4 is a relatively small cardiomyocyte population and many cells are expressing Nrf1 at high levels. Do the seemingly 2 populations observed in this scatter plot correspond to CM2 and CM4?

Response: Thank you for this suggestion. We have now included a plot to show the averaged expression of Nrf1 in the CM1-CM5 populations in **revised Supplementary Fig. 4c**. This plot clearly depicts the enriched Nrf1 expression in the CM4 population. As you pointed out that CM4 is a relatively small population, we plotted the expression of Nrf1 and Acta2 in individual cardiomyocytes for all populations in Fig. 1n. This result shows that the expression of Nrf1 highly correlates with that of Acta2, which is a marker gene of CM4. It is very likely that the secondary branch of cells in Fig. 1n corresponds to CM2 cells, given that Acta2 is also enriched in CM2 cells, albeit less abundantly compared to CM4 cells (**Fig. 1a**).

2. In Figure 2A and subsequent experiments. What is the rationale for performing MI at P3, compared to P1 MI performed in previous figures?

Response: Thank you for this question. In Figure 2, we studied the effect of Nrf1 deletion on neonatal heart regeneration using the aMHC-Cre:Nrf1^{fl/fl} mice we created. These mice are maintained in the C57/BL6 background, which, in our experience, is frequently associated with maternal cannibalization when MI surgery was performed at P1. This observation is consistent with our previous report¹³. We found that maternal cannibalization can be significantly reduced if we perform MI at a later stage such as P3. From our previous studies, we know that the ability of the heart to regenerate gradually diminishes during the first week after birth. At P3, the WT heart still possesses a significant ability to regenerate after MI as we showed in Fig. 2b, therefore allowing us to study the effect of Nrf1 deletion at this time point.

3. Figure 2h-i. This reviewer understands that the number of mice on which Figure 2h-I is based was 4-6 per genotype (and experimental group; i.e. sham/MI?). Hearts are pooled for the analysis, leaving Figure 2i subject to the presence of outliers.

Response: You are correct. We pooled dissected hearts from 4-6 mice for each group (WT-Sham, WT-MI, Nrf1KO-Sham, and Nrf1KO-MI), with the reasoning that this normalizes any potential dissection and individual biases. We agree that there could potentially be outliers in our samples. However, this potential bias is more likely to be 'diluted out' when multiple hearts are pooled for analyses than if they were individually processed.

- Extended data Figure 5a, which complements data Fig 2h-i shows a low number of CMs in WT-sham (n=1,406) compared to cKO-sham (n=3,328). While percentage numbers seem

similar, what is the potential effect of this for the detection of lower abundance CM populations? For example, CM2 accounts for perhaps 3% of CM, which is some 40 cells.

Response: While we aimed to target the same number of cells for all samples, the final number of cells being sequenced is often hard to control precisely. This is due to many factors, such as nuclear quality and accuracy of counting, all of which are difficult to precisely control. However, we are confident that our data represents the accurate composition of cardiomyocytes in the heart, given that the fraction of each cardiomyocyte population in the WT-Sham heart analyzed in this study, albeit with less sequenced nuclei, is highly consistent with an independently performed snRNA-seq from our previous report (Cui et al. 2020).

- The axis is split in two segments in this panel. Please split the bars accordingly to avoid misinterpretations. Regarding this panel, the authors write “after MI, the Nrf1 cKO hearts showed a significant increase of CM5 cells”. This seems visually clear, but it is not clear how this significance is achieved on a pooled sample?

Response: Thank you for the suggestion. We have now split the bar graphs in Fig. 2i. We also performed Chi-squared tests on cell abundance between MI and Sham samples. This analysis showed that the increase of CM5 cells in the MI samples is statistically significant compared to the Sham samples. We have included this information in the **revised Fig. 2i** and its corresponding figure caption.

- Last, while Extended data Figure 5a-b shows a similar median number of genes per cell when comparing ShamWT vs shamKO (and in MI), the distribution shown in panel 5b displays obvious differences in top percentiles. How does this affect data quality and interpretation?

Response: As we described in Methods, we normalized gene expression measurements of each cell by its total expression counts. To account for variation in the number of genes and UMI detected in each cell, we normalized the expression matrix using the ScaleData function and set the vars.to.regress to nUMI and nGenes. This approach has been extensively applied in single-cell studies to control for these types of variations, which, as we know, are difficult to control experimentally and are intrinsic to single-cell sequencing approaches in general.

4. Better descriptions are needed for Supplementary Tables, including n-numbers. Regarding Supp Table 1, which complements Fig2L, the number of genes and GO terms identified would benefit from a side comparison of CM4 cell populations before MI. A full table for the enriched GO terms presented in Fig2L would be required, and it should include fold-enrichments and number of genes affected (i.e. in background gene list and in altered genes).

Response: Thank you for these helpful suggestions. We have now added detailed descriptions of Supplemental Data Tables, including numbers of differentially expressed genes and annotations of each column. As suggested, we also included a table to show GO terms enriched for upregulated and down-regulated genes with p-values and gene names. These new tables can be found in the **revised Supplementary Data 1** and are shown under tabs: ‘Nrf1KO upregulated GO terms’ and ‘Nrf1KO down-regulated GO terms.’

5. While it is relevant for CM4, the increase in ROS levels shown in Fig2M cannot be attributed (or immediately linked) to the observations made in CM4 cells as this may be a general effect, specially considering that CM4 are a relatively minor CM population. Have the authors check this in other CM cells?

Response: Because we do not currently have tools to delete Nrf1 specifically in CM4 cells, in this study we deleted Nrf1 throughout all cardiomyocytes. Given that Nrf1 is also expressed in other CMs albeit at lower levels (Fig 1 o-q and the revised Supplementary Fig. 3c), it is likely that Nrf1 also plays a protective role in these CMs in response to injury. Thus, the increased ROS levels in Nrf1 cKO hearts could be a general effect, which further supports the important stress-response function of Nrf1 in the heart. To check ROS levels in other cardiomyocyte populations, we would need to have a robust approach to isolate different cardiomyocytes as intact viable cells, which is impossible with our current single-nucleus sequencing approach. Nevertheless, the emphasis of our study is that Nrf1 expression is required for neonatal heart regeneration, as our data in Fig. 2 have shown.

6. Data presented in Figure 3 is remarkable, and the positive effect of Nrf1 delivery seems clear. However, a few questions arise from this figure:

- Do adult LV cardiomyocytes split into different populations? Although they will be terminally differentiated, the benefits of Nrf1 delivery might be associated to particular adult CM populations. The authors do not perform additional snRNAseq in adults.

Response: In this study, we identified the protective factor Nrf1 by its enriched expression in regenerative cardiomyocytes (CM4). Because CM4 cells disappear in non-regenerative hearts, we wanted to test whether grossly overexpressing Nrf1 in adult cardiomyocytes could reactivate the regenerative state and elicit a protective effect against ischemic injury.

Several recent studies have reported that adult murine and human cardiomyocytes are heterogeneous populations^{14, 15}. Therefore, it is possible that a subset of cardiomyocytes in the adult heart is preferentially targeted for Nrf1-mediated protection and is responsible for the improved cardiac function. We agree with the reviewer that additional snRNAseq could be useful to delineate the population-specific effect. However, we think that a transgenic mouse model that uniformly overexpresses Nrf1 in all cardiomyocytes would be the most ideal for this analysis, as the AAV delivery approach we used in this experiment seldom achieves 100% transduction efficiency. This mosaic nature of AAV-mediated gene expression could potentially create artificial heterogeneity in cardiomyocytes attributed to Nrf1 expression level, which could in turn affect the downstream data analysis. We are currently making a Tetracycline-inducible transgenic mouse line that can activate Nrf1 expression under the control of aMHC-tTA. We will use this mouse model to comprehensively characterize the protective effect of Nrf1 *in vivo*, including any potential differential response between different cardiomyocyte populations, as suggested by the reviewer.

- AAV injection is provided at postnatal day P4, but I/R experiments are performed when mice are 8 weeks old. This is important. AAV vectors will have a great tropism for post-mitotic cells and, at P4, will not reach proliferative cardiomyocytes (i.e. CM4) with a great avidity. Given the time at which Nrf1 is provided, it is plausible that some CM populations with no proliferative capacity at the time of delivery will be contributing to the observed effect. It would be interesting to see which CM populations uptake the vector. For this, perhaps a few key markers for each population would suffice. This would better link the experiments performed in adults with those performed in neonates in Figure 2.

Response: It is well known that AAV can infect both dividing and nondividing cells *in vivo*¹⁶, thus it is unlikely that CM4 cardiomyocytes are being disadvantageously infected at the time of

AAV-Nrf1 delivery. Additionally, in our study, AAV was systemically delivered at P4. It usually takes several days for AAV to turn on gene expression in the heart. Given that CM4 is a transient population, it would be technically challenging to distinguish viral uptake in cells that are CM4-derived versus other CMs without lineage tracing. Regardless, we want to point out that we were not trying to overexpress Nrf1 in any specific cardiomyocyte population, but rather in all cardiomyocytes to achieve maximal protection in the heart.

- As mice will have non-physiological levels of Nrf1 for 8 weeks until I/R is induced, can the authors elaborate on why they chose P4 as the best time for AAV delivery?

Response: We chose P4 as the time point for AAV delivery for practical reasons. First, systemic AAV delivery at P4 results in consistent and efficient viral transduction in the heart and is a procedure we routinely perform in the lab. Second, we have not been able to effectively deliver the AAV virus systemically in adult mice. Due to the substantial increase of body weight in adult mice, we would need to drastically scale up the viral production in order to obtain the large amount of virus needed to achieve the same dosage (5E13 viral genome/kg of body weight) used in this study. Currently, our lab does not possess the capacity for large scale AAV production nor is the core facility that we use capable of producing high titer virus within the time limit allowed for this revision. Importantly, we want to point out that even with prolonged Nrf1 expression, these mice did not show any baseline difference in cardiac function (Fig. 3 e and f), precluding any potential effect of Nrf1 overexpression independent of injury.

- Does AAV9 delivery improve the outcome in Nrf1-cKO mice?

Response: Our results showed that Nrf1 cKO and wildtype mice have comparable levels of cardiac function in the Sham condition (Fig. 2b and Supplementary Fig. 5b), therefore it is unlikely that AAV9-Nrf1 will further improve the performance of the heart at the baseline level. However, given that Nrf1 is a stress-coping factor, it is plausible that AAV9-Nrf1 can improve heart function in Nrf1 cKO mice after injury, as in wildtype mice that we show in Fig. 3. To test that, we would need to deliver the AAV9 virus to Nrf1 cKO mice followed by ischemic injury in adulthood. These experiments are estimated to take at least six months to finish, which is the time allowed for this revision. Most importantly, in our view, the results from these experiments do not substantially add to the information we have already shown in the current manuscript, that is Nrf1 overexpression confers cardioprotection in the heart against ischemic injury.

7. Although not as pronounced, Nfr2 confers a remarkable level of protection as shown in Figure 3K. What is the role of Nfr2 or even Nfr3 (not expressed in P14 hearts) in Nrf1-cKOs? As the authors nicely show, Nrf1 has unique roles in regulating the proteasome. Do any of the Nfe2 family genes exhibit compensatory effects, at least in physiological conditions, for the roles on which they overlap (i.e. regulation of RedOx balance)?

Response: We have shown that, among Nfe2 family members, only Nrf1 and Nrf2 are expressed in the heart (Supplementary Fig. 4a). To study the possible redundant function between Nrf1 and Nrf2, we generated the cardiomyocyte-specific knockout of Nrf2 as well as Nrf1 plus Nrf2 (double knockout) and compared them to Nrf1 cKO mice. We saw that Nrf1 cKO reduced the expression of analyzed proteasome and antioxidant target genes to greater degrees than Nrf2 cKO. Furthermore, the reduction of these target genes by Nrf1 deletion was comparable to Nrf1/Nrf2 dKO, suggesting that Nrf1 plays a dominant role in regulating the expression of these genes in the heart. We have included these results in the **new Supplementary Fig. 11**.

8. Extended Data Figure 10. Why do the authors use log₂ peptide counts in a proteomics analysis involving TMT labelling? Looking at the methods provided, the addition of TMT does not seem obvious to me. I would expect a quantification based on reporter ion intensities (i.e. 6-plex TMT with two groups of 3 samples each). Please revise.

Response: You are correct. The plotted values in Extended Data Figure 10 (now Supplementary Fig. 12) are reporter ion intensities, not peptide counts. We have corrected the figure legend and Methods in the revised manuscript (page 25) to reflect that. Thank you for picking up this error!

9. Calcium transient experiment in Figure 5n is based on n=29-33 cells per group. How many times was this experiment repeated? This also applies to panels 5o and 5p, based on the same experiment.

Response: The experiments shown in Fig. 5n-p and s were independently repeated twice with different batches of hPSC-CMs. We have now revised the figure legend to specify that.

10. Figures 1F to 1M are based on single heart sections. Can the authors show additional images from different hearts to demonstrate reproducibility?

Response: Thank you for the suggestion. We have now generated spatial transcriptome data from a different heart collected at 7-days post-MI. Our analysis showed consistent results between the two replicates (**new Supplementary Fig. 1**). Additionally, we thought spatial transcriptome analysis on Sham hearts at a different time point after MI would be informative, so we also performed this analysis on heart sections collected at 3- and 7- days post-Sham as well 3-days post-MI surgery. These results are depicted in the **new Supplementary Fig. 1 and Supplementary Fig. 2** and revealed dynamic localization of cardiac cell types during heart regeneration.

Minor.

1. Can the authors comment on the impact of CM bi-nucleation in snRNAseq experiments? This is more relevant as CMs exit the earliest postnatal stage and a higher percentage become binucleated.

Response: We agree with you that some nuclei in our snRNAseq data are likely to be from the same cardiomyocyte due to binucleation, especially in later postnatal stages, such as P8, when the majority of cardiomyocytes are binucleated. This potentially affects our estimation of percentages of different cardiomyocyte populations. For example, the CM1 population comprises mature cardiomyocytes that are more likely binucleated, thus it is possible that we overestimated its abundance by counting nuclear frequency. Unfortunately, this is a general problem for all single-nuclear RNA-seq studies and is a trade-off for being able to sequence transcripts of single cardiomyocytes that would otherwise be difficult to capture due to their large size. Importantly, cardiomyocyte binucleation arises from incomplete cell division instead of cell fusion, thus the two nuclei from the same cardiomyocytes should share the same gene expression profile. This makes it unlikely to affect the transcriptomic analyses we performed in our study for identification of cell types and differential gene expression.

2. Please spell out abbreviation of Nrfl in the abstract.

Response: Thank you for this suggestion. We have now included the full gene name of Nrfl in the abstract: "... found that they are marked by expression of Nrfl, a stress responsive

transcription factor encoded by the Nuclear Factor Erythroid 2 Like 1 (*Nfe2l1*) gene.”

Reviewer #4 (Remarks to the Author):

Cui et al in their ms: Mechanistic interplay between cardioprotection and heart regeneration mediated by Nrf1, explore this role of Nrf1 in infarcted neonatal mouse heart. A dual stress response mechanism is described involving activation of the proteasome and maintenance of redox balance. A mechanistic interplay between adaptive stress responses and heart regeneration is described and highlight the central role of Nrf1 in these processes. These findings are truly original and are based on the combination of state of the art methods.

Response: Thanks for your support and enthusiasm!

Figure 1 shows the results from the spatial transcriptomics analysis. It shows that CM4 is upregulated one day after LAD ligation and this is verified with Acta2 RNA-scope analysis. The figure shows nicely where the infarction scar area is located in the HE image of the heart. Spatial distribution of the projected snRNA-seq defined celltypes is shown individually and also as pie charts at every data point with the proportion of the different cell types. Although the distributions look logical, no description is given of which snRNA-seq markers genes were used for the different cell types (Only generally as ref 7).

Response: Thank you for your suggestion. We have now included spatial expression of marker genes for each cell type in **new Supplementary Fig. 1c**. We also added more detailed description in Results as well as Methods to better explain how our analysis was performed (**pages 5 and 29**).

Also, no validation is presented on the accuracy of the proportions of cell types in each data point. Typically, in each data point there should be 10-15 cells. Given the size of cardiomyocytes it might be fewer cells.

Response: We agree with you that validation of cell types for each data point is important. However, in order to do that, the same tissue section used for spatial sequencing would also need to be analyzed either using in situ hybridization or antibody staining to label different cell types. Unfortunately, the current protocol for spatial transcriptome profiling using the Visium platform by 10x Genomics has not been extensively tested or optimized to include these additional steps. Although this technical hurdle prohibited us from systematically validating the proportion of each cell type, using RNA-scope we showed that around 20-30% of cardiomyocytes express high levels of CM4 marker genes Acta2 and Nrf1 (Fig. 1c and q). This observed percentage of CM4 cells is consistent with our spatial-seq and snRNA-seq data, providing validation for our analyses.

Another unclarity is the use of snRNA data, that is single nuclear data for the deconvolution of cell types. Cardiomyocytes typically are polynuclear, that infer that there are less cardiomyocytes than cardiomyocyte nuclei in the data points. One uncertainty in this respect is if the different cardiomyocyte types have different degrees of polynucleation.

Response: This is another good point, related to the **minor point 1 raised by reviewer #3**. Please see our response above. In addition, we also want to add that our study focuses on the transcriptome and spatial localization of different CM populations instead of the degree of their

nucleation, which, albeit is interesting, is out of the scope for our study and would require different tools and approaches, such as ¹⁷ to address.

Further, it is questionable that some pie charts show only proportions of 5-10% for some cell types. Although the overall results seem feasible, the projection of snRNA seq cell type profiles is not trivial and would need a more precise and validated description.

Response: Thank you for your suggestion. We have now revised the Result and Method to include more information and better description of our analyses (pages 5 and 29).

Christer Sylvén

References:

1. Hortells, L., Johansen, A.K.Z. & Yutzey, K.E. Cardiac Fibroblasts and the Extracellular Matrix in Regenerative and Nonregenerative Hearts. *J Cardiovasc Dev Dis* **6** (2019).
2. Bassat, E. *et al.* The extracellular matrix protein agrin promotes heart regeneration in mice. *Nature* **547**, 179-184 (2017).
3. Notari, M. *et al.* The local microenvironment limits the regenerative potential of the mouse neonatal heart. *Science advances* **4**, eaao5553 (2018).
4. Karhu, S.T. *et al.* GATA4-targeted compound exhibits cardioprotective actions against doxorubicin-induced toxicity in vitro and in vivo: establishment of a chronic cardiotoxicity model using human iPSC-derived cardiomyocytes. *Arch Toxicol* **94**, 2113-2130 (2020).
5. Porrello, E.R. *et al.* Transient regenerative potential of the neonatal mouse heart. *Science (New York, N.Y.)* **331**, 1078-1080 (2011).
6. Sereti, K.I. *et al.* Analysis of cardiomyocyte clonal expansion during mouse heart development and injury. *Nature communications* **9**, 754 (2018).
7. He, L. *et al.* Proliferation tracing reveals regional hepatocyte generation in liver homeostasis and repair. *Science (New York, N.Y.)* **371** (2021).
8. Kretzschmar, K. *et al.* Profiling proliferative cells and their progeny in damaged murine hearts. *Proceedings of the National Academy of Sciences of the United States of America* **115**, E12245-e12254 (2018).
9. Klochendler, A. *et al.* A transgenic mouse marking live replicating cells reveals in vivo transcriptional program of proliferation. *Developmental cell* **23**, 681-690 (2012).
10. Hirotsu, Y., Hataya, N., Katsuoka, F. & Yamamoto, M. NF-E2-related factor 1 (Nrf1) serves as a novel regulator of hepatic lipid metabolism through regulation of the Lipin1 and PGC-1 β genes. *Molecular and cellular biology* **32**, 2760-2770 (2012).
11. Hirotsu, Y. *et al.* Transcription factor NF-E2-related factor 1 impairs glucose metabolism in mice. *Genes Cells* **19**, 650-665 (2014).
12. Wang, Z. *et al.* Mechanistic basis of neonatal heart regeneration revealed by transcriptome and histone modification profiling. *Proceedings of the National Academy of Sciences of the United States of America* **116**, 18455-18465 (2019).
13. Xin, M. *et al.* Hippo pathway effector Yap promotes cardiac regeneration. *Proceedings of the National Academy of Sciences of the United States of America* **110**, 13839-13844 (2013).

14. Gladka, M.M. *et al.* Single-Cell Sequencing of the Healthy and Diseased Heart Reveals Cytoskeleton-Associated Protein 4 as a New Modulator of Fibroblasts Activation. *Circulation* **138**, 166-180 (2018).
15. Litviňuková, M. *et al.* Cells of the adult human heart. *Nature* **588**, 466-472 (2020).
16. Daya, S. & Berns, K.I. Gene therapy using adeno-associated virus vectors. *Clin Microbiol Rev* **21**, 583-593 (2008).
17. Kannan, S. *et al.* Large Particle Fluorescence-Activated Cell Sorting Enables High-Quality Single-Cell RNA Sequencing and Functional Analysis of Adult Cardiomyocytes. *Circulation research* **125**, 567-569 (2019).

REVIEWER COMMENTS

Reviewer #1 (Remarks to the Author):

The author addressed all my comments.

The only remark I do have: are the authors sure to measure 20S proteasome (in absence of ATP both 20S and 26S contribute to the activity).

Better to call it: ATP-non-stimulable proteasomal activity.

Reviewer #2 (Remarks to the Author):

The authors made a remarkable job in revising their manuscript according to the reviewer's recommendations, and should be commended for this. I am fully satisfied with the revision and with to congratulate the authors for their excellent work.

Reviewer #3 (Remarks to the Author):

This is a revised version of a previously submitted by Cui et al. The authors have addressed my initial concerns with additional data or new experiments. The addition of Supplementary Fig. 11 (double KO model) to test the possibility of Nrf1-Nrf2 redundancy is appreciated, as it is the re-formatting of the supplemental tables. Other questions have been appropriately addressed by amending the manuscript. This reviewer also understands that the focus of the manuscript is not on the CM4 population of cardiomyocytes, but on the role of Nrf1. As such, I appreciate that the discussion separates clearly the results on neonates from those obtained in adult mice in which Nrf1 is overexpressed.

Only a few concerns remain, which relate to my previous comments:

Major (requiring experiments):

1. Regarding question 6. “As mice will have non-physiological levels of Nrf1 for 8 weeks until I/R is induced, can the authors elaborate on why they chose P4 as the best time for AAV delivery?”

The authors reason that “these mice did not show any baseline difference in cardiac function (Fig. 3 e and f), precluding any potential effect of Nrf1 overexpression independent of injury”. The authors only test for differences in cardiac function, but they need to exclude other effects at baseline (pre-MI). For example, sustained Nrf1 overexpression could lead to changes in the proportion of cardiac cells including fibroblasts and resident inflammatory cells, without necessarily affecting baseline cardiac function. In my opinion, a characterization of a few hearts from AAV-Nrf1-treated vs. AAV-Tomato-treated mice at baseline would be advisable.

Concerns not requiring further experiments:

1. With regards to my second question in the original submission: What is the rationale for performing MI at P3, compared to P1 MI performed in previous figures?

The authors justify this decision in the fact that maternal cannibalization is observed in C57/Bl6 mice when MI is performed at P1. This is a reasonable response but, if that is the case, it could be reasoned that experiments in Figure 1 should have been done in animals at P3 instead. It is important that the assumptions taken from Figure 1 in terms of cell populations hold true in subsequent experiments. Data in Figure 1 is in part a re-purposing / re-analysis of a previous paper. While repeating an experiment of this magnitude would be too demanding, this should at least be mentioned as a potential limitation in the discussion.

2. Regarding my third question: “Extended data Figure 5a, what is the potential effect of this [low overall cardiomyocyte counts] for the detection of lower abundance CM populations? For example, CM2 accounts for perhaps 3% of CM, which is some 40 cells.”

The authors respond that “This is due to many factors, such as nuclear quality and accuracy of counting, all of which are difficult to precisely control”. This is not critical for the paper overall, as not much focus is put on the low abundant CM2 population. However, if accuracy of counting is not well controlled, any claims made for CM2 (lines 334-337 in the revised manuscript) should be accompanied by an acknowledgement of this limitation.

3. Regarding my previous question: “Does AAV9 delivery improve the outcome in Nrf1-cKO mice?”

The authors respond that “To test that, we would need to deliver the AAV9 virus to Nrf1 cKO mice followed by ischemic injury in adulthood. These experiments are estimated to take at least six

months to finish, which is the time allowed for this revision.” This reviewer understands the complexity of adding those experiments. However, I would advise to explicitly mention in the discussion that future rescue experiments would further prove the claims.

4. I wondered when I read the first version of this manuscript about the impact of binucleation, especially in the context of this paper, using hearts at different maturation stages. I believe this inherent limitation of the single-nuclear RNAseq technology should be mentioned in the discussion.

Reviewer #4 (Remarks to the Author):

The concerns I have raised have been responded in a balanced and adequate manner. I have no further comments.

REVIEWER COMMENTS

Reviewer #1 (Remarks to the Author):

The author addressed all my comments.

The only remark I do have: are the authors sure to measure 20S proteasome (in absence of ATP both 20S and 26S contribute to the activity).

Better to call it: ATP-non-stimulable proteasomal activity.

Response: We thank the reviewer for this suggestion. We modified the method to specify that it was the ATP-non-stimulable proteasomal activity that we measured in our study (page 25). We also amended the text and figures in our revised manuscript. It now consistently states the proteasome' activity (instead of just the 20S proteasome) throughout the Figures and Figure Legends.

Reviewer #2 (Remarks to the Author):

The authors made a remarkable job in revising their manuscript according to the reviewer's recommendations, and should be commended for this. I am fully satisfied with the revision and with to congratulate the authors for their excellent work.

Response: We thank this reviewer for the comments and the enthusiasm for our work.

Reviewer #3 (Remarks to the Author):

This is a revised version of a previously submitted by Cui et al. The authors have addressed my initial concerns with additional data or new experiments. The addition of Supplementary Fig. 11 (double KO model) to test the possibility of Nrf1-Nrf2 redundancy is appreciated, as it is the re-formatting of the supplemental tables. Other questions have been appropriately addressed by amending the manuscript. This reviewer also understands that the focus of the manuscript is not on the CM4 population of cardiomyocytes, but on the role of Nrf1. As such, I appreciate that the discussion separates clearly the results on neonates from those obtained in adult mice in which Nrf1 is overexpressed.

Only a few concerns remain, which relate to my previous comments:

Major (requiring experiments):

1. Regarding question 6. "As mice will have non-physiological levels of Nrf1 for 8 weeks until I/R is induced, can the authors elaborate on why they chose P4 as the best time for AAV delivery?" The authors reason that "these mice did not show any baseline difference in cardiac function (Fig. 3 e and f), precluding any potential effect of Nrf1 overexpression independent of injury". The authors only test for differences in cardiac function, but they need to exclude other effects at baseline (pre-MI). For example, sustained Nrf1 overexpression could lead to changes in the proportion of cardiac cells including fibroblasts and resident inflammatory cells, without necessarily affecting baseline cardiac function. In my opinion, a characterization of a few hearts from AAV-Nrf1-treated vs. AAV-Tomato-treated mice at baseline would be advisable.

Response: As requested by the reviewer, we have now performed immunostaining on sections of uninjured hearts collected from AAV-Nrf1-treated and AAV-Tomato-treated mice. We labeled

cardiac fibroblasts (Vimentin+), resident immune cells (CD45+), and smooth muscle cells (Acta2+) as well as quantifying their frequency in duplicates. Our results showed that these cell types are present in comparable abundance in AAV-Nrf1 and AAV-Tomato hearts. These results are shown in the revised Supplementary Figure 7. We also updated the main text on page 9: “At the baseline level, AAV-Nrf1 hearts had similar cardiac function and proportions of cardiac fibroblasts as well as tissue-resident immune cells compared to AAV9- TdTomato hearts (Supplementary Fig. 7b-e)”.

Concerns not requiring further experiments:

1. With regards to my second question in the original submission: What is the rationale for performing MI at P3, compared to P1 MI performed in previous figures?

The authors justify this decision in the fact that maternal cannibalization is observed in C57/Bl6 mice when MI is performed at P1. This is a reasonable response but, if that is the case, it could be reasoned that experiments in Figure 1 should have been done in animals at P3 instead. It is important that the assumptions taken from Figure 1 in terms of cell populations hold true in subsequent experiments. Data in Figure 1 is in part a re-purposing / re-analysis of a previous paper. While repeating an experiment of this magnitude would be too demanding, this should at least be mentioned as a potential limitation in the discussion.

Response: We appreciate the thorough reasoning of this reviewer. As the reviewer pointed out, the spatial transcriptome profiling shown in Figure 1 was a parallel analysis to the snRNA-seq study from our previous paper, aiming to visualize the different cardiac cell types during heart regeneration. To keep the timepoints consistent, we performed MI at P1. Although our later MI experiments on Nrf1 cKO hearts were performed at P3, we do not think that this difference is a limitation of our study. First, in our previous paper, we showed that the cardiomyocyte composition between P1 and P4 sham hearts did not drastically change (CM1-5 populations were all identified in both stages at comparable abundance). Second, we used the well-established label-transferring approach implemented in Seurat v3 to identify cell types in P1 injured hearts and P3 injured hearts using the same query dataset generated from our previous paper (see Method, page 30 and 33-34). Third, the comparison of cell populations and gene expression was done within samples from the same injury timepoints. For example, Figure 1, Supplementary Fig 1, and Supplementary Figure 2 showed the localization of different cardiac cell types all from P1 MI hearts. Similarly, Figure 2 compared the differential gene expression in CM4 cells between wildtype and Nrf1cKO hearts only at P3 MI. Therefore, we are confident about the cell type identification in P1 MI and P3 MI hearts as well as the subsequent analyses for each timepoint.

2. Regarding my third question: “Extended data Figure 5a, what is the potential effect of this [low overall cardiomyocyte counts] for the detection of lower abundance CM populations? For example, CM2 accounts for perhaps 3% of CM, which is some 40 cells.”

The authors respond that “This is due to many factors, such as nuclear quality and accuracy of counting, all of which are difficult to precisely control”. This is not critical for the paper overall, as not much focus is put on the low abundant CM2 population. However, if accuracy of counting is not well controlled, any claims made for CM2 (lines 334-337 in the revised manuscript) should be accompanied by an acknowledgement of this limitation.

Response: We thank the reviewer for this suggestion. We have now modified the text to acknowledge that the CM2 cells were detected at a low frequency (page 16): “Although CM2 cells

are detected at a low frequency (~3%), it is possible that they are localized more broadly, accounting for the observed BrdU+ cardiomyocytes in remote zones”.

We also like to add that it is the number of input cells/nuclei and the final captured cells/nuclei that cannot be precisely controlled, which is common to all single-cell sequencing studies. However, as we stated in our first response, we are confident that our data represents the accurate **composition** of cardiomyocyte nuclei in the heart, given that the fraction of each cardiomyocyte population analyzed in this study is highly consistent with our previous report (Cui et al. 2020).

3. Regarding my previous question: “Does AAV9 delivery improve the outcome in Nrf1-cKO mice?”

The authors respond that “To test that, we would need to deliver the AAV9 virus to Nrf1 cKO mice followed by ischemic injury in adulthood. These experiments are estimated to take at least six months to finish, which is the time allowed for this revision.” This reviewer understands the complexity of adding those experiments. However, I would advise to explicitly mention in the discussion that future rescue experiments would further prove the claims.

Response: We thank the reviewer for this suggestion. We have now included a sentence in the Discussion to talk about the recommended rescue experiments (page 17):
“Future rescue experiments by overexpressing Nrf1 and its target genes in Nrf1 cKO hearts will establish the role of Nrf1 in maintaining this regeneration permissive condition and further delineate the functional downstream pathways.”

4. I wondered when I read the first version of this manuscript about the impact of binucleation, especially in the context of this paper, using hearts at different maturation stages. I believe this inherent limitation of the single-nuclear RNAseq technology should be mentioned in the discussion.

Response: We thank the reviewer for this suggestion. We have now mentioned this technical limitation of snRNA-seq in the Discussion (page 16):
“Additionally, one limitation of snRNA-seq is its inability to distinguish mono-nucleated versus binucleated cardiomyocytes. This inherent technical limitation potentially affects our estimation of percentages of different cardiomyocyte populations especially in later postnatal stages such as P8, when the majority of cardiomyocytes become binucleated. Future experiments using large particle fluorescence-activated cell sorting will reveal any potential differences in the degree of their nucleation. Nevertheless, the majority of cardiomyocyte binucleation arises from incomplete cell division instead of cell fusion, thus the two nuclei from the same cardiomyocytes should in principle share the same gene expression profile, which makes it unlikely to affect the transcriptomic analyses performed in this study.”

Reviewer #4 (Remarks to the Author):

The concerns I have raised have been responded in a balanced and adequate manner. I have no further comments.

Response: We thank this reviewer for the comment.

REVIEWERS' COMMENTS

Reviewer #3 (Remarks to the Author):

No further comments. I thank the editors for their detailed response.